# The influence of aftershocks on seismic hazard analysis: A case study from Xichang and the surrounding areas

Qing Wu[1], Guijuan Lai[1], Jian Wu[2], Jinmeng Bi[1,3]

[1]Institute of Geophysics, China Earthquake Administration, Beijing, China
[2]China Earthquake Disaster Prevention Center, Beijing, China
[3]Tianjin Earthquake Agency, Tianjin, China

*Correspondence to*: Qing Wu (wuqing908@sina.com), Guijuan Lai (laigj@cea-igp.ac.cn)

**Abstract**: In some instances, a strong aftershock can cause more damage than the mainshock. Ignoring the influence of aftershocks may underestimate the seismic hazard of some areas. Taking Xichang and its surrounding areas as an example, and based on the Seismic Ground Motion Parameter Zoning Map of China (GB18306-2015), this study used the Monte Carlo method to simulate synthetic mainshock sequences. Additionally, the Omi-Reasenberg-Jones (Omi-R-J) aftershock activity model is used to simulate the aftershock sequences that follow mainshocks above a certain magnitude threshold. Then, the mainshock and the aftershocks are combined to calculate the regional seismic hazard using ground motion prediction equations (GMPEs). Finally, the influence of aftershocks on seismic hazard analysis is examined and considered. The results show that in areas with moderate to strong seismic backgrounds, the influence of aftershocks on probabilistic seismic hazard analysis can exceed 50%. These results suggest that the impact of aftershocks should be properly considered for future probabilistic seismic hazard analyses, especially in areas with moderate to strong seismic activity backgrounds and in areas prone to secondary disasters such as landslides and mudslides.

**Keywords**: Aftershocks; Omi-R-J model; ETAS model; Monte Carlo method; Seismic hazard analysis.

## 1 Introduction

Aftershocks are commonly removed from observed earthquake catalogs during probabilistic seismic hazard analyses, assuming that mainshocks follow a Poisson distribution. However, the strong aftershocks that follow an earthquake may cause more damage than the mainshock and should not be underestimated. As there is not enough time to repair damage between the mainshock and subsequent aftershocks,

buildings can suffer cumulative damage from aftershocks, which can lead to additional casualties and property losses (Bi et al., 2022). For example, after the 1976 M7.8 Tangshan earthquake in China, most houses in the aftershock zone collapsed, and the railway lines on the deck of a local bridge were damaged during the M7.1 and M6.9 aftershocks (Lv et al., 2007). The M5.0 aftershock that followed the 2003 M8.0 Hokkaido earthquake in Japan caused a secondary fire disaster due to a spilled tank (Zhao et al., 2005). The M6.3 aftershock that followed the 2010 M7.1 Christchurch earthquake in New Zealand caused damage to buildings, 146 deaths, and over 300 people to go missing (Zhang et al., 2011). Lv et al. (2007) statistically analyzed the aftershocks that followed 21 M >7.0 mainshocks in China and found that the proportion of peak ground accelerations caused by aftershocks that exceeded that of the mainshock was 76.2%; that is, aftershocks may cause more severe damage than the mainshock. Therefore, ignoring the impact of aftershocks may underestimate the seismic risk in some areas. The cumulative damage-induced losses caused by strong aftershocks have attracted considerable attention in the field of disaster and catastrophe insurance modeling (Xiong, 2019).

Cornell (1968) proposed the classical probabilistic seismic hazard analysis (PSHA) method, and based on that work, Wiemer (2000) proposed the aftershock probabilistic seismic hazard analysis (APSHA). Gallovič & Brokešová (2008) combined the generalized form of the Omori law (Omori, 1894; Utsu, 1961; Utsu et al, 1995) that was given by Shcherbakov et al. (2004), refined the APSHA steps and parameterizations, and analyzed the seismic hazard probability of aftershocks that followed several earthquakes as case studies. Shen & Yang (2018) used the APSHA method established by Gallovič & Brokešová (2008) to analyze the aftershock seismic hazard probability of the 2017 M7.0 Jiuzhaigou earthquake in China. In addition, many scholars have derived the influence of aftershocks on seismic hazard analysis by using analytical solutions (Yeo & Cornell, 2009; Marzocchi & taroni, 2014; Iervolino et al., 2014; Davoudi et al., 2020; Taroni & Akinci, 2021). Boyd (2012) and Xu & Wu (2017) used the Epidemic-type Aftershock Sequence (ETAS, Ogata, 1988,1998) model to generate catalogs with and without aftershocks. They used a spatially smooth seismicity model to calculate the impact of aftershock clusters for

probabilistic seismic hazard analysis. Canales & Baan (2020) used the Poisson model to generate mainshock sequences and the ETAS model to generate aftershock sequences. Field et al. (2021) use the Third Uniform California Earthquake Rupture Forecast (UCERF3) ETAS model (UCERF3-ETAS) to evaluate the effects of declustering and Poisson assumptions on seismic hazard estimates. Wang et al. (2022) compared the ETAS-simulated hazard with approximations based on the declustered Poisson approach (DP), the nondeclustered Poisson approach (NDP), and the recently proposed sequence-based PSHA (Iervolino et al., 2014).

Based on the Seismic Ground Motion Parameter Zoning Map of China (GB18306-2015), this study used the Monte Carlo method to simulate synthetic mainshock sequences. Then, according to the magnitude of the mainshock, the Omi-Reasenberg-Jones (Omi-R-J) aftershock activity model (Omi, 2013, 2016, 2019) is used to simulate the aftershock sequences that follow mainshocks for a certain magnitude threshold. Finally, the mainshocks and the aftershocks are combined to calculate the regional seismic hazard using ground motion prediction equations (GMPEs). Thus, the influence of aftershocks on seismic hazard analysis is analyzed.

Xichang city, one of the three major space launch facilities in China, is located in the Anning River Valley in southwestern Sichuan Province. The Anning River fault and the Zemu River fault run through the city. Historically, three M $\geqslant$ 7.0 earthquakes have occured in the region: a M7.0 event in 814, a M7.5 earthquake in 1536 and a M7.5 event in 1850 (Fig. 1). The Anning River fault is one of the main faults in the North-South Sichuan-Yunnan tectonic belt and is also an important fault in Southwest China. According to regional geological data (Li, 1993; He & Ikeda, 2007), the Anning River fault zone is the boundary of different tectonic units with Paleozoic to Mesozoic ages. The west side of the fault contains a metamorphic complex and magmatic rock belt, and a Mesozoic-Cenozoic sedimentary basin lies on the east side. The Zemu River fault has been active throughout the Holocene (Li, 1997; Du, 2000) and is connected to the Anning River fault zone in the north and the Xiaojiang fault zone in the south. The fault has an overall strike of 330°, a fault plane dip angle of more than 60° and a dip direction of northeast or southwest. Since the late Quaternary, the Anning River fault and Zemu River fault have been characterized

by continuous strike-slip movements (Xu et al., 2003a; Xu et al., 2003b). The Anning River fault and Zemu River fault are located at the boundary of the central Yunnan secondary block in the rhombus-shaped Sichuan-Yunnan block, which controls the focal positions of most nearby earthquakes with M ≥ 7 (Lu et al., 2012).

Xichang is located in an area prone to strong earthquakes. Considering the impact of aftershocks in seismic hazard assessment, it is of critical importance to focus on fortifying areas subject to strong aftershocks, especially against landslides, debris flows and other secondary geological disasters. However, these preparations require the development of accurate disaster prevention technologies.

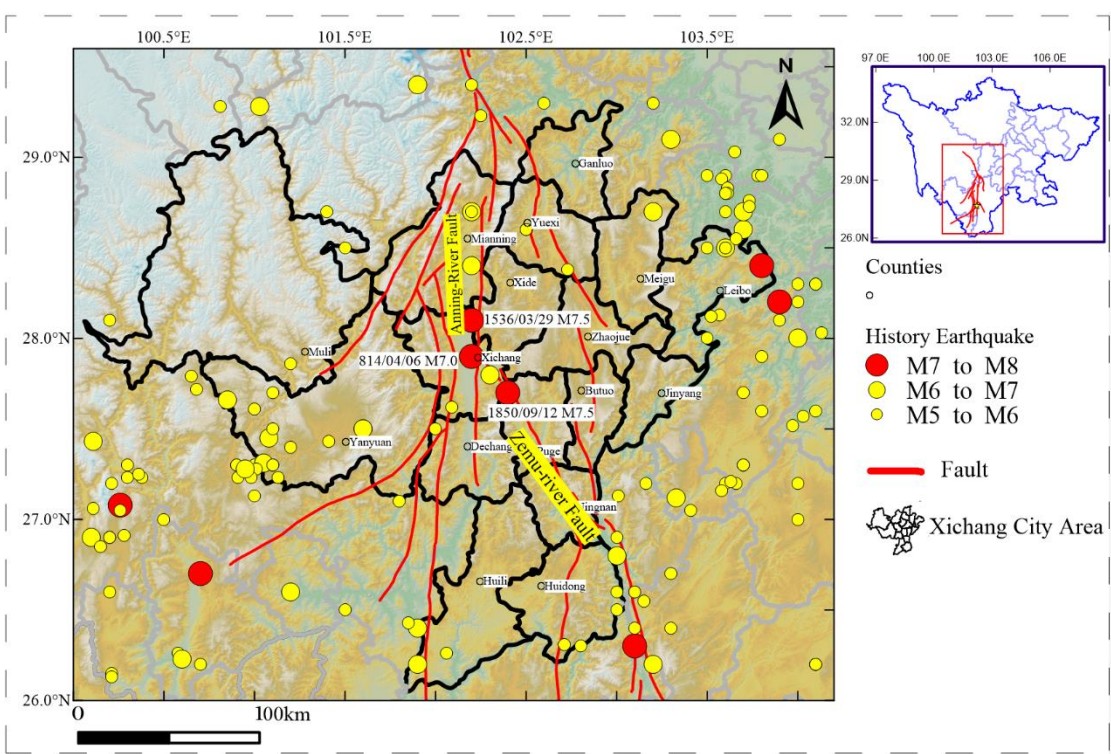

**Figure 1 Distribution of seismic events and the tectonic background in Xichang and its surrounding areas**

## 2. Aftershock activity models and their parameters

### 2.1 Omi-R-J aftershock sequence model

After moderate or strong earthquakes, when direct information is available, the early activity characteristics of the aftershock sequences are used for sequence type determination (Jiang et al., 2007), strong aftershock prediction (Omi et al., 2013) and short-term aftershock probability prediction (Reasenberg & Jones, 1989; Gerstenberger et al., 2005). These characteristics have important scientific value and

practical significance in earthquake relief, regional earthquake risk assessment and understanding of the earthquake sequence itself. Reasenberg and Jones (1989) developed the R-J model to predict the occurrence rate of early aftershocks based on the Omori-Utsu formula (Omori 1894; Utsu 1961) and the Gutenberg-Richard (G-R) law (Gutenberg & Richard, 1944).

According to the R-J model, the aftershock intensity function, with a magnitude no less than M at time $t$ in the earthquake sequence, can be expressed as:

$$\lambda(t, M) = \frac{k}{(t+c)^p} 10^{-bM} \tag{1}$$

where $t$ is the time after the mainshock. The parameter $k$ controls the overall aftershock productivity. The parameter $p$ represents the decay degree of the seismic sequence. Parameter $c$ is used to adjust the incompleteness of the aftershock records within a very short time after the mainshock. This parameter is a positive and small constant and is negatively correlated with the focal depth (Shebalin & Narteau, 2017). Parameter $b$ represents the stress accumulation level (Wiemer & Katsumata, 1999; Enescu et al., 2011). This model is simple in principle and suitable for estimating the parameters of moderate to strong earthquake sequences with simple decay laws. As a classical seismic sequence analysis method, it is widely used in aftershock prediction throughout the world and for earthquake hazard assessments by the Global Earthquake Model (GEM) project.

After the occurrence of moderate or strong earthquakes, many small aftershocks will be "submerged" in the early stage, resulting in a reduction in the completeness of the earthquake catalogs, making it difficult to assess many of the small earthquakes below the magnitude of completeness. Based on the R-J model, Omi et al. (2013) proposed the "Omi-R-J" model by considering the aftershocks below the magnitude of completeness during the early stage of the earthquake sequence in the model parameter fitting and in the aftershock occurrence rate prediction. Omi et al. (2013) used the expression of the detection rate function $q$(M) given by Ogata and Katsura (1993) (OK1993 model) to describe the detection rate of the incomplete part of the earthquake catalog. The actual recorded earthquake probability density function can be expressed as:

$$P(M|\beta,\mu,\sigma) = \frac{e^{-\beta M} q(M|\mu,\sigma)}{\int_{-\infty}^{+\infty} e^{-\beta M} q(M|\mu,\sigma) dM} = \frac{e^{-\beta M} q(M|\mu,\sigma)}{e^{(-\beta\mu+\beta^2\sigma^2/2)}/\beta} = \beta e^{-\beta(M-\mu)+\beta^2\sigma^2/2} q(M|\mu,\sigma) \qquad (2)$$

where $\beta$ is equal to $b\ln10$, $\mu$ represents the corresponding magnitude when the detection rate is 50%, $\sigma$ is the corresponding magnitude dispersion and $\mu+2\sigma$ or $\mu+3\sigma$ is usually used to approximate the minimum magnitude of completeness $M_c$. In the parameter estimation of formula (2), the "state-space" model developed by Omi et al. (2013) was used to estimate the time varying factor $\mu(t)$. Specifically, $\mu(t)$ is set as the discrete distribution function corresponding to the aftershock time sequence $t_i \leqslant t \leqslant t_{i+1}$ ($i=1$、$2……n$). The hyperparameter $V$ is set to control the smoothness of the distribution, assuming a priori distribution with a smooth constraint on $\mu(t)$. After the parameters $\beta$, $\sigma$, and $V$ are optimized and the maximum a posteriori estimation is performed by the maximum expectation (EM) iterative algorithm, the parameter $\boldsymbol{\mu}=(\mu_1, \mu_2……\mu_n)^T$ is obtained by Bayesian estimation. In the early period after a mainshock, the waveforms of small earthquakes are submerged by the waveforms of large earthquakes, making it difficult to identify small earthquakes and resulting in the lack of a catalog of small earthquakes. The EM algorithm is based on the super parameter estimation of the Newton iterative algorithm. It can optimize the parameters in the case of missing small earthquakes in the early period, reducing the error of the Newton iterative algorithm and obtaining more objective parameters. The earthquake detection rate function considering incomplete earthquake records can be expressed as $v(t,M) = \lambda(t,M)q(M|\mu(t),\sigma)$. The logarithmic likelihood function related to parameters $p$, $c$, and $k$ is:

$$\ln L(k,c,p) = \sum_{M_i \geq M_c} \ln v(t,M) - \int_{M_c}^{\infty} dM \int_0^T dt v(t,M) \qquad (3)$$

where $t_i$ and $M_i$ are the time and magnitude of the $i$-th aftershock that occurred within the "learning period" [0, T] during model fitting. Then, the parameters $p$, $c$, $k$ and the standard deviation are determined by combining the Omori-Utsu formula and the maximum likelihood method.

## 2.2 ETAS time series model

The ETAS model introduces the idea of self-similarity and assumes that both background earthquakes and triggered earthquakes can stimulate their own aftershocks, and many direct aftershocks and indirect aftershocks (aftershocks of

aftershocks) can be generated after a mainshock. Therefore, the ETAS model is constructed with branch point process characteristics (Ogata, 1988; Bi & Jiang, 2019). The conditional intensity function can be expressed as:

$$\lambda(t) = \mu_{ETAS} + K_{ETAS} \sum_{t_i < t} \frac{e^{\alpha_{ETAS}(M_i - M_0)}}{(t - t_i + c_{ETAS})^{p_{ETAS}}}, M_i > M_0 \qquad (4)$$

where $t\text{-}t_i$ represents the elapsed time of seismic event $i$ and $K_{ETAS}$ is a normalized constant that determines the expected number of aftershocks directly triggered by the $M_i$ event. The parameter $\alpha_{ETAS}$ represents the ability of a seismic event to stimulate secondary aftershocks (Ogata 1989; 1992). Compared with isolated earthquakes and main aftershocks, the $\alpha_{ETAS}$ of a swarm-type earthquake sequence is smaller, generally $\alpha_{ETAS} < 1$ (Ogata, 2001), and $p_{ETAS}$ represents the decay degree of the seismic sequence. Parameter $c_{ETAS}$ is used to adjust the incompleteness of aftershock records within a very short time following the mainshock. Parameter $\mu_{ETAS}$ indicates the occurrence rate of background earthquakes. In the calculation process, when the occurrence rate of background earthquakes in the area is low, $\mu_{ETAS}=0$ is used to better ensure the stability of parameter fitting.

The maximum likelihood method (MLEs) is used to estimate the parameters [$K_{ETAS}$, $c_{ETAS}$, $\alpha_{ETAS}$, $p_{ETAS}$] in the ETAS model. The likelihood function L is expressed as:

$$log\,L = \sum_{i:S \le t_i < T} lg\,\lambda(\,t_i\,) - \int_S^T \lambda(\,t\,)dt \qquad (5)$$

## 2.3 Aftershock sequence models in the Xichang area

Gao (2015) divided the Chinese mainland and its adjacent areas into 29 seismic belts, of which 25 seismic belts are located inside mainland China. Since 1970, the Xianshuihe East-Yunnan seismic belt, where the Xichang area is located (see Fig. 2), has experienced six M ≥ 7.0 earthquakes; the 1970 M7.8 Tonghai earthquake in Yunnan, 1973 M7.6 Luhuo earthquake in Sichuan, 1973 M7.2 Nima earthquake in Tibet, 1974 M7.1 Daguan earthquake in Yunnan, 1997 M7.4 Nima earthquake in Tibet, and 2010 M7.1 Yushu earthquake in Qinghai. As the early seismic monitoring ability in Tibet is limited and the number of recorded aftershocks is low, the two M7.0+ earthquakes in Tibet cannot be used to fit aftershock parameters. We estimated the aftershock sequence parameters of the other 4 M7.0+ earthquakes by using the ETAS

model and Omi-R-J model. The results are shown in Table 1 and Table 2. To obtain

more samples of aftershock sequence parameters, we use the Omi-R-J model to

calculate the aftershock sequence parameters of 40 M4.5-7.0 earthquakes. The results

are also shown in Table 2.

**Table 1 Basic information on four mainshocks with M ≥ 7.0 and aftershock sequence parameters, as calculated by the ETAS model, from the Xianshuihe East-Yunnan seismic belt, where the Xichang area is located**

| No. | 1 | 2 | 3 | 4 |
|---|---|---|---|---|
| Time (BJT) | 1970/01/05 01:00:34 | 1973/02/06 18:37:05 | 1974/05/11 03:25:16 | 2010/04/14 07:49:36 |
| Longitude | 102.6 | 100.4 | 104 | 96.59 |
| Latitude | 24.1 | 31.5 | 28.1 | 33.22 |
| Magnitude | 7.8 | 7.6 | 7.1 | 7.3 |
| p | 1.27 | 1.01 | 0.96 | 0.99 |
| Error (p) | 0.14 | 0.11 | 0.07 | 0.05 |
| c | 0.0323 | 0.0235 | 0.0053 | 0.0016 |
| Error (c) | 0.0327 | 0.0214 | 0.0109 | 0.0014 |
| k | 0.0276 | 0.0284 | 0.002 | 0.036 |
| Error (k) | 0.0162 | 0.0178 | 0.0022 | 0.0089 |
| α | 1.51 | 1.22 | 2.06 | 0.96 |
| Error (α) | 0.24 | 0.17 | 0.28 | 0.11 |
| $M_C$ | 3.3 | 2 | 2 | 2 |
| $C_0$ | 0.1024 | 0.0742 | 0.0491 | 0.0186 |
| Number of events above $M_C$ | 364 | 585 | 728 | 461 |
| Number of events | 1278 | 1044 | 947 | 2558 |

**Table 2 Basic information on 44 mainshocks with M ≥ 4.5 and aftershock sequence parameters, as calculated by the Omi-R-J model, from the Xianshuihe East-Yunnan seismic belt, where the Xichang area is located**

| No. | Time (BJT) | Lon. | Lat. | Mag. | p | Error(p) | c | Error(c) | k | Error(k) | b | Error(b) |
|---|---|---|---|---|---|---|---|---|---|---|---|---|
| 1 | 1970/01/05 01:00:34.34 | 102.6 | 24.1 | 7.8 | 1.34 | 0.05 | 0.3212 | 0.0553 | 0.0231 | 0.012 | 0.84 | 0.05 |
| 2 | 1973/02/06 18:37:05.05 | 100.4 | 31.5 | 7.6 | 0.95 | 0.04 | 0.1524 | 0.0515 | 0.0004 | 0.0002 | 0.92 | 0.04 |
| 3 | 1974/05/11 03:25:16.16 | 104 | 28.1 | 7.1 | 0.86 | 0.02 | 0.0204 | 0.0084 | 0.0085 | 0.0033 | 0.78 | 0.03 |
| 4 | 2010/04/14 07:49:36.36 | 96.59 | 33.22 | 7.3 | 0.81 | 0.01 | 0.0041 | 0.002 | 0.0052 | 0.0014 | 0.71 | 0.02 |
| 5 | 1970/07/31 21:10:46.46 | 103.6 | 28.53 | 5.4 | 0.88 | 0.09 | 0.0403 | 0.0557 | 0.0103 | 0.0117 | 0.99 | 0.12 |
| 6 | 1971/08/16 12:57:59.59 | 103.6 | 28.8 | 5.9 | 1.11 | 0.06 | 0.9602 | 0.2154 | 0.3468 | 0.1227 | 0.7 | 0.04 |
| 7 | 1972/09/30 04:24:39.39 | 101.57 | 30.17 | 5.7 | 0.69 | 0.06 | 0.007 | 0.016 | 0.0057 | 0.0058 | 0.77 | 0.09 |
| 8 | 1975/01/12 05:22:27.27 | 101.53 | 24.8 | 5.4 | 0.67 | 0.04 | 0.0223 | 0.0309 | 0.0553 | 0.0212 | 0.67 | 0.03 |
| 9 | 1975/01/15 19:34:37.37 | 101.8 | 29.43 | 6.2 | 0.99 | 0.07 | 0.0651 | 0.0312 | 0.0128 | 0.0107 | 0.81 | 0.06 |
| 10 | 1975/07/09 21:55:42.42 | 103.03 | 23.88 | 5.3 | 0.59 | 0.05 | 0.0045 | 0.0099 | 0.0041 | 0.0051 | 0.82 | 0.06 |
| 11 | 1976/11/07 02:04:05.05 | 101.08 | 27.5 | 6.7 | 0.69 | 0.02 | 0.004 | 0.0034 | 0.0195 | 0.0071 | 0.83 | 0.03 |
| 12 | 1976/12/13 14:36:55.55 | 101.05 | 27.35 | 6.4 | 0.75 | 0.05 | 0.0214 | 0.0263 | 0.0087 | 0.0063 | 0.85 | 0.06 |
| 13 | 1978/05/20 09:40:52.52 | 100.3 | 25.55 | 5.3 | 0.62 | 0.03 | 0.03 | 0.0401 | 0.0385 | 0.0084 | 0.84 | 0.02 |
| 14 | 1978/09/26 05:49:36.36 | 99.58 | 29.87 | 5 | 0.51 | 0.06 | 0.0043 | 0.0127 | 0.0203 | 0.0143 | 0.88 | 0.09 |
| 15 | 1980/02/02 | 101.29 | 27.85 | 5.8 | 0.61 | 0.02 | 0.0036 | 0.0018 | 0.0242 | 0.0054 | 0.87 | 0.02 |

| # | Date | | | | | | | | | | | |
|---|---|---|---|---|---|---|---|---|---|---|---|---|
| | 20:29:14.14 | | | | | | | | | | | |
| 16 | 1982/06/16 07:24:32.32 | 100.03 | 31.96 | 6 | 1.12 | 0.03 | 0.0144 | 0.0037 | 0.0005 | 0.0006 | 1.06 | 0.07 |
| 17 | 1982/07/03 16:13:31.31 | 99.87 | 26.53 | 5.4 | 0.8 | 0.02 | 0.0138 | 0.006 | 0.0328 | 0.0109 | 0.83 | 0.03 |
| 18 | 1983/06/04 17:34:41.41 | 103.4 | 26.97 | 5 | 0.83 | 0.07 | 0.0012 | 0.0024 | 0.0185 | 0.0195 | 0.68 | 0.10 |
| 19 | 2001/02/23 08:09:20.20 | 101.1 | 29.42 | 6 | 0.92 | 0.1 | 0.0202 | 0.043 | 0.002 | 0.0052 | 0.93 | 0.15 |
| 20 | 2003/06/17 22:46:18.18 | 102.3 | 27.87 | 4.6 | 0.91 | 0.1 | 0.0248 | 0.0312 | 0.001 | 0.0011 | 1.03 | 0.10 |
| 21 | 2003/07/21 23:16:00.00 | 101.2 | 26 | 6.2 | 0.87 | 0.05 | 0.0182 | 0.0179 | 0.014 | 0.0088 | 0.84 | 0.06 |
| 22 | 2003/10/16 20:28:04.04 | 101.3 | 25.92 | 6.1 | 0.76 | 0.06 | 0.0238 | 0.0175 | 0.0076 | 0.0069 | 0.9 | 0.10 |
| 23 | 2003/11/15 02:49:43.43 | 103.7 | 27.2 | 5.1 | 0.54 | 0.04 | 0.0031 | 0.0076 | 0.0477 | 0.0211 | 0.63 | 0.04 |
| 24 | 2005/08/05 22:14:43.43 | 103.1 | 26.6 | 5.4 | 1.15 | 0.08 | 0.0999 | 0.0548 | 0.0045 | 0.0047 | 0.96 | 0.08 |
| 25 | 2008/08/30 16:30:52.52 | 101.92 | 26.28 | 6.1 | 1.05 | 0.08 | 0.0307 | 0.0287 | 0.017 | 0.0124 | 0.75 | 0.06 |
| 26 | 2009/07/09 19:19:14.14 | 101.03 | 25.6 | 6.3 | 1.14 | 0.06 | 0.1662 | 0.0633 | 0.0137 | 0.0063 | 0.78 | 0.03 |
| 27 | 2010/02/25 12:56:51.51 | 101.94 | 25.42 | 5.2 | 0.99 | 0.05 | 0.0019 | 0.0015 | 0.0051 | 0.003 | 0.79 | 0.07 |
| 28 | 2012/06/24 15:59:34.34 | 100.69 | 27.71 | 5.7 | 1.14 | 0.09 | 0.244 | 0.0489 | 0.0299 | 0.0206 | 0.96 | 0.02 |
| 29 | 2012/09/07 11:19:41.41 | 103.97 | 27.51 | 5.7 | 0.7 | 0.02 | 0.0043 | 0.0027 | 0.0606 | 0.0097 | 0.7 | 0.02 |
| 30 | 2013/01/18 20:42:50.50 | 99.4 | 30.95 | 5.5 | 1.2 | 0.08 | 0.0091 | 0.0059 | 0.0043 | 0.0058 | 0.86 | 0.10 |
| 31 | 2013/08/31 08:04:17.17 | 99.35 | 28.15 | 5.9 | 0.81 | 0.01 | 0.0028 | 0.0015 | 0.0156 | 0.0022 | 0.98 | 0.01 |
| 32 | 2014/01/15 03:17:46.46 | 101.17 | 26.86 | 4.5 | 0.99 | 0.09 | 0.0032 | 0.0136 | 0.0048 | 0.0082 | 0.74 | 0.11 |
| 33 | 2014/05/07 22:11:42.42 | 101.92 | 25.49 | 4.7 | 0.88 | 0.09 | 0.015 | 0.0194 | 0.0053 | 0.0069 | 0.86 | 0.13 |
| 34 | 2014/08/03 16:30:12.12 | 103.33 | 27.11 | 6.6 | 0.73 | 0.02 | 0.0047 | 0.0029 | 0.0264 | 0.0075 | 0.72 | 0.02 |
| 35 | 2014/08/17 06:07:59.59 | 103.51 | 28.12 | 5.2 | 0.84 | 0.07 | 0.011 | 0.0298 | 0.0258 | 0.0137 | 0.75 | 0.05 |
| 36 | 2014/10/01 09:23:29.29 | 102.74 | 28.38 | 5.2 | 1.14 | 0.1 | 0.1486 | 0.0976 | 0.0104 | 0.0083 | 0.78 | 0.07 |
| 37 | 2014/11/22 16:55:28.28 | 101.68 | 30.29 | 6.4 | 0.53 | 0.01 | 0.0043 | 0.0054 | 0.0004 | 0.0001 | 1.03 | 0.02 |
| 38 | 2016/09/23 00:47:13.13 | 99.6 | 30.08 | 5.2 | 1.01 | 0.03 | 0.0192 | 0.0073 | 0.0214 | 0.0095 | 0.92 | 0.05 |
| 39 | 2017/02/08 19:11:39.39 | 103.37 | 27.09 | 4.9 | 0.84 | 0.1 | 0.0359 | 0.058 | 0.0035 | 0.0042 | 0.96 | 0.13 |
| 40 | 2018/05/16 16:46:12.12 | 102.31 | 29.23 | 4.5 | 1.16 | 0.05 | 0.0023 | 0.0011 | 0.0138 | 0.0072 | 0.82 | 0.07 |
| 41 | 2018/08/13 01:44:25.25 | 102.72 | 24.18 | 5.1 | 1.14 | 0.05 | 0.1091 | 0.0315 | 0.1725 | 0.0418 | 0.61 | 0.02 |
| 42 | 2018/10/17 13:29:19.19 | 102.25 | 25.89 | 4.6 | 0.85 | 0.12 | 0.0305 | 0.143 | 0.0035 | 0.005 | 0.94 | 0.11 |
| 43 | 2018/10/31 16:29:56.56 | 102.09 | 27.62 | 5.1 | 1.4 | 0.1 | 1.2964 | 0.3135 | 0.0006 | 0.0008 | 1.06 | 0.06 |
| 44 | 2018/12/13 23:32:52.52 | 98.84 | 29.6 | 4.9 | 1.08 | 0.07 | 0.009 | 0.0061 | 0.0192 | 0.0179 | 0.83 | 0.11 |

# 3 Probabilistic seismic hazard analysis method considering aftershocks

## 3.1 Probabilistic seismic hazard analysis considering only the Poisson distribution

Wu et al. (2020) used the Monte Carlo method to simulate synthetic earthquake catalogs for probabilistic seismic hazard analysis based on the Seismic Ground Motion Parameters Zonation Map of China (GB18306-2015). The seismic source zone model used by the Seismic Ground Motion Parameters Zonation Map of China

(GB18306-2015) is based on seismological and geological data for China. To reflect the heterogeneity in potential seismicity and describe the structural complexity more faithfully, the model adopts a three-level delineation of seismic belts, uses background and structural sources, and considers the tectonic differences between eastern and western China (Zhou et al. 2013). The spatial relationship of the three source levels is as follows (see Fig. 2): the base layer is the seismic belt (seismic statistical area), which is used to reflect the overall statistical characteristics of seismicity; the middle layer is the background potential sources, which are used to reflect the differences in seismic characteristics of small- and moderate-magnitude earthquakes under different tectonic conditions; and the upper layer consists of the structural potential sources, which are used to reflect the small-scale spatial seismic heterogeneity caused by the differences in local seismic structural conditions. This is a peculiar property of the seismicity model used for probabilistic seismic hazard assessment in China (CPSHA). Figure 2 shows the spatial distribution of the potential sources for the Xianshuihe East-Yunnan seismic belt where the Xichang area is located.

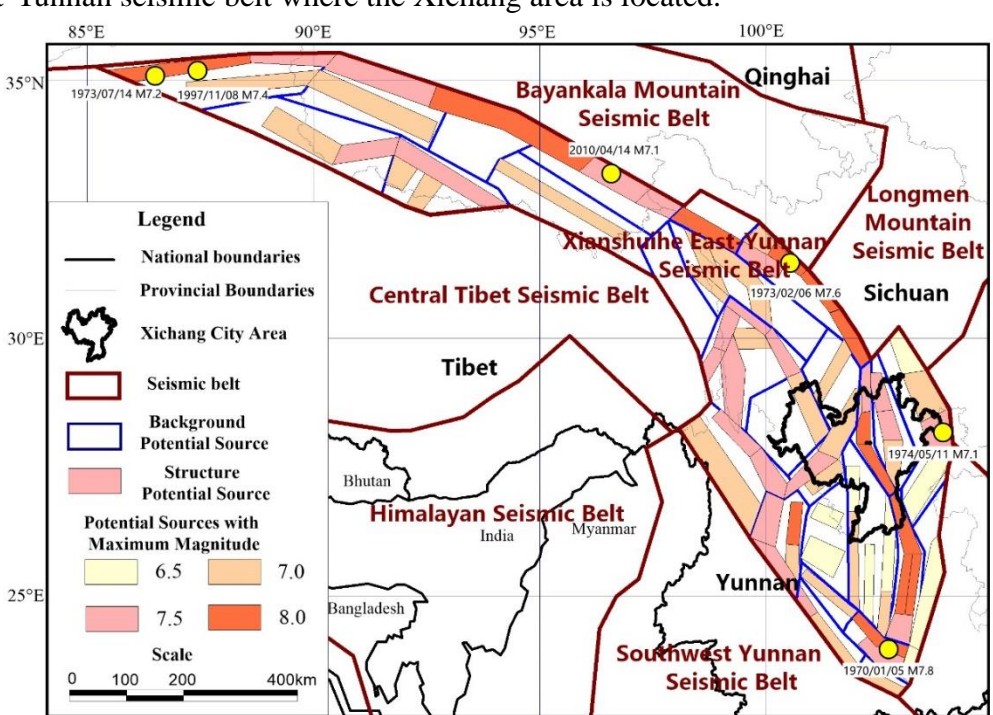

**Figure 2 The spatial distribution of the potential sources for the Xianshuihe East-Yunnan seismic belt where the Xichang area is located, and six M ≥ 7.0 earthquakes in the belt since 1970**

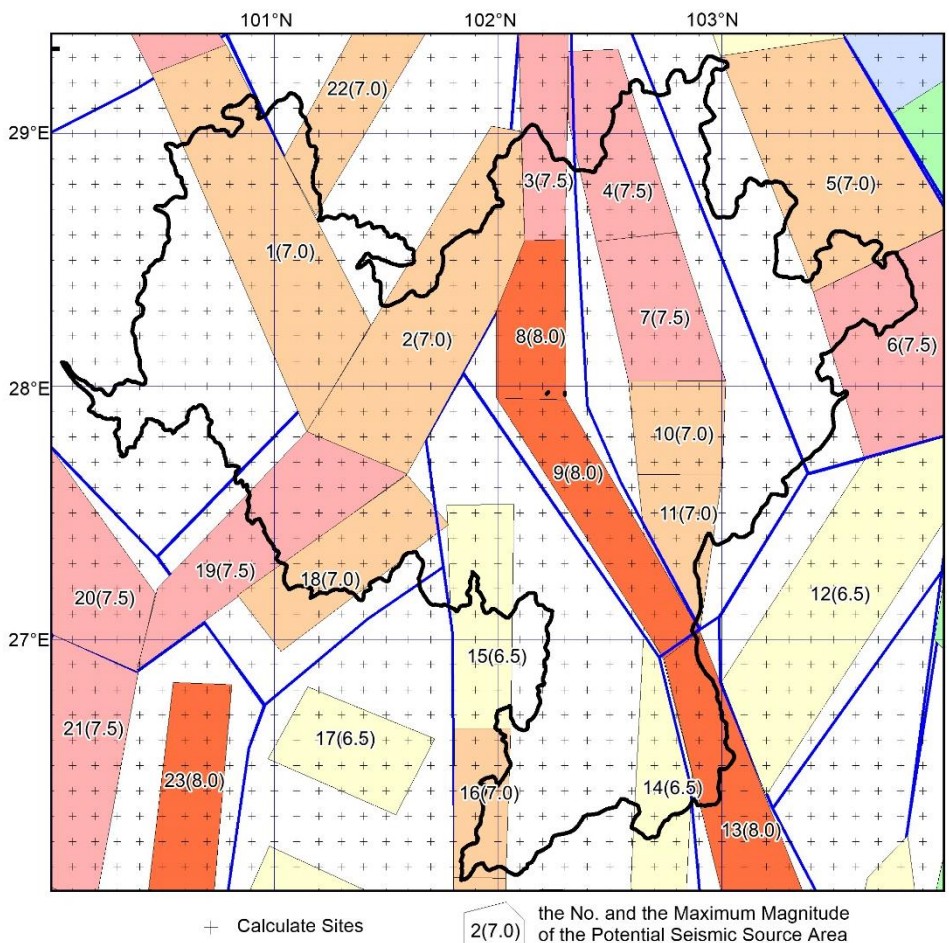

**Figure 3 The calculation sites and the potential seismic sources in and around Xichang. The meaning of the color of the potential seismic source is the same as that in Figure 2.**

The seismic zoning map of China (GB18306-2015) has established the corresponding probability model and spatial distribution model of earthquake occurrence, and gives the basic parameters of each seismic zone. Figure 3 shows the potential seismic sources in and around Xichang.

According to the basic assumptions and seismicity parameters of the zoning map (Table 3), the following steps are used to synthesize the sets of earthquake sequences (Wu & Gao, 2018; Wu et al., 2020):

(1) Based on the assumption that the occurrence of earthquakes in seismic zones satisfies the Poisson distribution, the time length T of the simulated earthquake sequence and the average annual occurrence rate $v_4$ of earthquakes with magnitude 4 and above in the seismic zone should be determined first. Then, a Poisson distribution random number $L$ is generated with T and $v_4$ as parameters, where $L$ is the number of earthquakes in the seismic zone for the length of time T to be simulated.

**Table 3 The list of seismicity parameters of potential seismic sources in and around Xichang**

| No. | $M_{UZ}$ | b value | $V_4$ | Strike | No. | $M_{UZ}$ | b value | $V_4$ | Strike |
|-----|----------|---------|-------|--------|-----|----------|---------|-------|--------|
| 1 | 7.0 | 0.85 | 32 | 120° | 13 | 8.0 | 0.85 | 32 | 120° |
| 2 | 7.0 | 0.85 | 32 | 55° | 14 | 6.5 | 0.85 | 32 | 80° |
| 3 | 7.5 | 0.85 | 32 | 90° | 15 | 6.5 | 0.85 | 32 | 90° |
| 4 | 7.5 | 0.85 | 32 | 115° | 16 | 7.0 | 0.85 | 32 | 90° |
| 5 | 7.0 | 0.85 | 32 | 120° | 17 | 6.5 | 0.85 | 32 | 150° |
| 6 | 7.5 | 0.85 | 32 | 120° | 18 | 7.0 | 0.85 | 32 | 30° |
| 7 | 7.5 | 0.85 | 32 | 115° | 19 | 7.5 | 0.85 | 32 | 45° |
| 8 | 8.0 | 0.85 | 32 | 90° | 20 | 7.5 | 0.85 | 32 | 120° |
| 9 | 8.0 | 0.85 | 32 | 125° | 21 | 7.5 | 0.85 | 32 | 55° |
| 10 | 7.0 | 0.85 | 32 | 90° | 22 | 7.0 | 0.85 | 32 | 55° |
| 11 | 7.0 | 0.85 | 32 | 80° | 23 | 8.0 | 0.85 | 32 | 75° |
| 12 | 6.5 | 0.85 | 32 | 50° | | | | | |

(2) Based on the assumption that the magnitude distribution of seismic zones satisfies the truncated Gutenberg-Richter relationship (magnitude-frequency relationship), with the minimum magnitude level $M_0$ and the maximum magnitude level $M_{UZ}$, the magnitudes of earthquakes to be simulated are determined.

The magnitude-frequency relationship is represented as:

$$\log N = a - bM \tag{6}$$

where $a$ and $b$ are coefficients, $N$ is the number of earthquakes whose magnitude are equal to or greater than $M$, and the initial magnitude of the zoning map is 4. The cumulative number of earthquake events is:

$$N(M) = e^{a-bM} \tag{7}$$

If we take $\Delta M = 0.1$, then

$$N(M) > N(M + \Delta M) \tag{8}$$

Based on $M$=4.1, 4.2, 4.3,…, $M_{UZ}$, a random number $u$ that satisfies a uniform distribution between 0 and 1 is generated. Then, the following is determined:

$$u \in \frac{N(M + \Delta M)}{N(4)} \sim \frac{N(M)}{N(4)} \tag{9}$$

If the above formula is true, the magnitude $M$ of an earthquake event is determined.

(3) Determination of epicenter location. First, the potential source area $H$ where the earthquake is located should be determined. According to the magnitude $M$ determined in the previous step, the magnitude range $d$ to which the earthquake

belongs is determined. Because the probability $P_d(\mathrm{h})$ of each magnitude range in each potential source area is known, a random number $u$ is generated that satisfies a uniform distribution between 0 and 1. The following is then determined:

$$u \in \sum_{h=1}^{H-1} P_d(h) \sim \sum_{h=1}^{H} P_d(h) \qquad (10)$$

If so, the potential source area $H$ where the earthquake event is located is

determined. Based on the assumption that the epicenter is evenly distributed in the potential source area, a point is randomly selected in the potential source area $H$ as the epicenter location of an earthquake.

(4) According to the azimuth of the potential source area, the azimuth of the earthquake is determined.

At this point in the calculation, the basic elements of an earthquake have been determined. Steps (2) ~ (4) are repeated until the required number $L$ of earthquakes in the seismic zone is obtained, accounting for all possible seismic zones that may affect the site, thus determining a seismic sequence and completing one sampling.

If the time length $T$ is set to one year, the seismic sequence obtained by one

sampling is called a one-year seismic sequence in this paper. When the time length is set to 10 years, the sequence is called a 10-year earthquake sequence.

For each earthquake in seismic sequences, the peak ground acceleration (PGA) of each site is calculated by the optimal ellipse search algorithm through ground motion prediction equations (GMPEs).

For 5000000 simulations of a 1-year earthquake sequence, if a site is affected by ground motions exceeding specific values, the sequence is assigned a value of 1. The sum of earthquake sequences identified as 1 is counted, and is divided by the total number of earthquake sequence simulations (i.e., 5000000), resulting in the annual exceedance probability of specific ground motions. Through the annual exceedance

probability, the 50-year exceedance probabilities of 10% and 2% can be calculated.

This is how the probabilities of seismic hazard are obtained.

## 3.2 Probabilistic seismic hazard analysis considering aftershocks

Since there are only a few strong earthquakes with M $\geqslant$ 7 in the Xianshuihe East-Yunnan seismic belt, the Omi-R-J model is selected as the aftershock parameter

model. According to the spatial division of the Seismic Ground Motion Parameters

Zonation Map of China (GB18306-2015), the median values of the $p$, $c$, $K$ and $b$ values (see Table 2) used in the Omi-R-J model (Omi et al., 2013, 2016, 2019) for the aftershock sequence samples from the Xianshuihe East-Yunnan seismic belt are 0.8747, 0.0187, 0.0133 and 0.8361, respectively. The aftershock sequences are generated according to these median values and the following steps:

(1) The mainshock sequences are simulated by the Monte Carlo method based on the Seismic Ground Motion Parameters Zonation Map of China (GB18306-2015). Each synthetic sequence represents a 1-year possible distribution of earthquakes in the region that is consistent with the seismicity model (Wu et al., 2020). Considering the destructiveness of the earthquake, when the magnitude threshold for the mainshock is met (i.g., M $\geqslant$ 6.0 in this study considering a potential sufficiently large impact on the site, and the value can be adjusted as needed), the aftershock sequence sampling is started.

(2) The minimum magnitude of the aftershock sequence is set to 4.0, and the maximum magnitude is equal to the magnitude of the mainshock. In fact, the magnitude of aftershocks can be greater than that of the mainshock. In this study, we focus on the 'aftershocks', so we adopted the assumption of Iervolino et al. (2014). That method assumes foreshocks do not contribute exceedances, aftershocks do not trigger their own aftershocks, and aftershocks are smaller than the mainshocks. The aftershock sequence satisfies the magnitude-frequency relationship $N(M)=10^{a-bM}$. The aftershock occurrence time $t$ is within 30 days after the mainshock and follows the Omori-Utsu formula $N(t)=\dfrac{K}{(t+c)^p}$. The time interval between a strong aftershock and the mainshock varies from a few seconds to several years, but most strong aftershocks occur a few days or even a day after the main shock (Japan Meteorological Agency, 2009; Tahir et al., 2012). A length of 30 days is taken as the duration for a simplified calculation, and can be changed as needed. According to the median values of $p$, $c$, $K$ and $b$ and the upper limit of magnitude of the potential sources, the magnitude and time series of aftershocks with M $\geqslant$ 4 are simulated.

(3) According to the empirical relationship between the magnitude of the mainshock and the rupture scale (Wells & Coppersmith, 1994), the rupture length and width are calculated by:

$$L=10^{(-3.22+0.69M_{\mathrm{W}})} \tag{11}$$

$$W=10^{(-1.01+0.32M_{\mathrm{W}})} \tag{12}$$

The rupture strike is taken as the direction of the potential source area where the mainshock is located, and the model of Felzer & Brodsky (2006) is adopted; that is, the aftershock density decays exponentially with increasing distance $r$ from the fault, $\rho(r) = cr^{-n}$, where $n$ is 1.37, and $c$ is a constant. Thus, the locations of the aftershock epicenters can be determined.

(4) The number of aftershocks. We have accounted for the number of M4.0+ aftershocks for M5.0+ mainshocks in the Chinese mainland and its surrounding area, and found that when the mainshock is greater than 6.0, the number of M4.0+ aftershocks within a month (30 days) increases with the magnitude of the mainshock, yielding the statistical relationship: log10 ($N$)=0.84$M$-4.57 (shown by

the red line in Fig. 4). This relationship fluctuates within the range of ±0.8 (shown by the two dotted red lines), and obeys the normal distribution under linear coordinates. The number of aftershocks corresponding to a certain magnitude is generated according to this relationship.

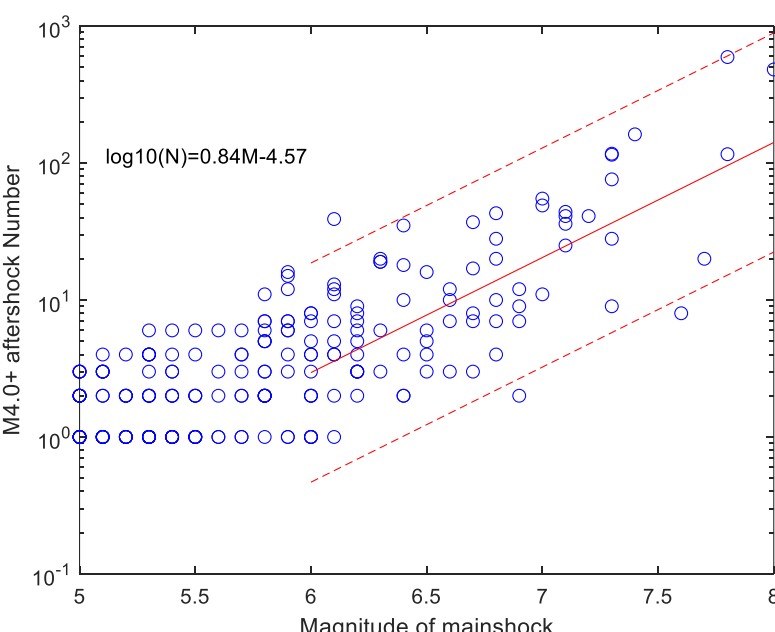

**Figure 4 The M5.0+ mainshocks and the number of their M4.0+ aftershocks for the Chinese**

**mainland and its surrounding areas.**

Figure 5 (a) shows a schematic diagram of the spatial distribution of 1-year mainshocks with M $\geq$ 6 sampled 100 times. The yellow star in the figure is the location of the mainshock. Figure 5 (b) is a schematic diagram of the spatial distribution of the corresponding aftershocks. The small blue dot in the figure is the

aftershock corresponding to the mainshock. The distribution direction of the aftershocks refers to the strike of the potential source area where the mainshock is located. In this study, considering the destructiveness of the earthquake, when the magnitude of the mainshock is $\geq$ 6.0, random sampling of the aftershock sequence begins, and the sampling time is set within 30 days after the mainshock. The model

program user interface can be used to adjust and refine the aftershock model to account for random aftershock sequences in the future that may have different requirements. To ensure the stability of the results, we conducted 5 million 1-year samplings.

(a) Mainshocks

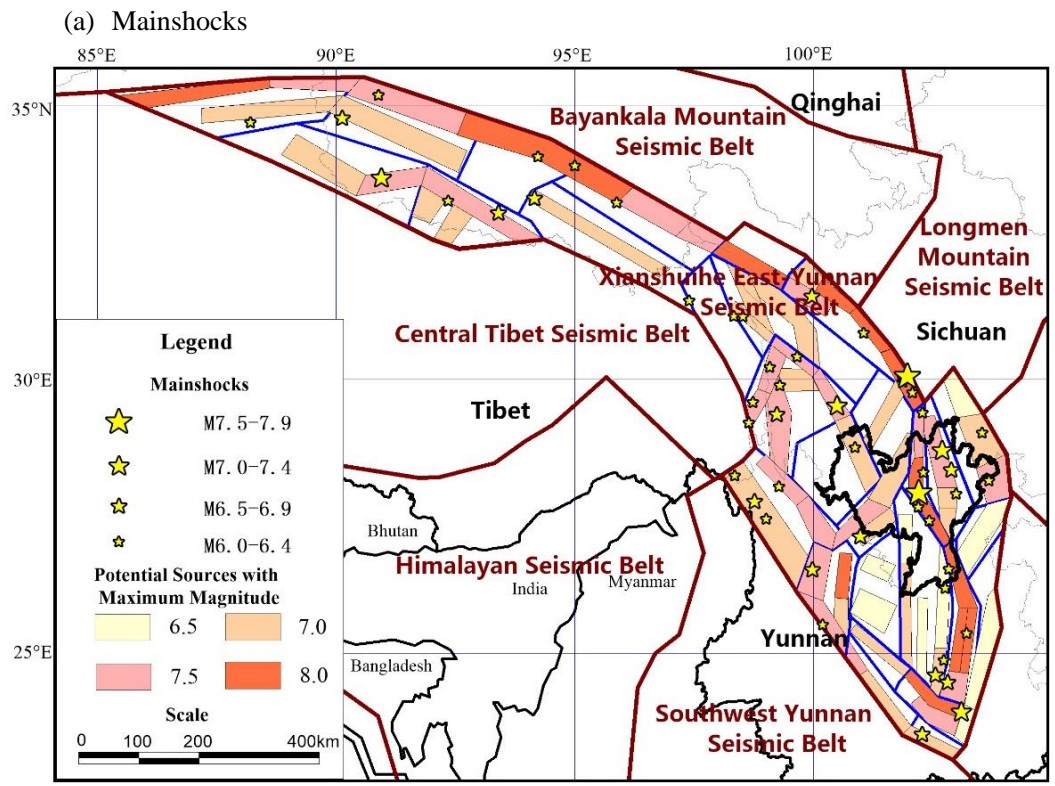

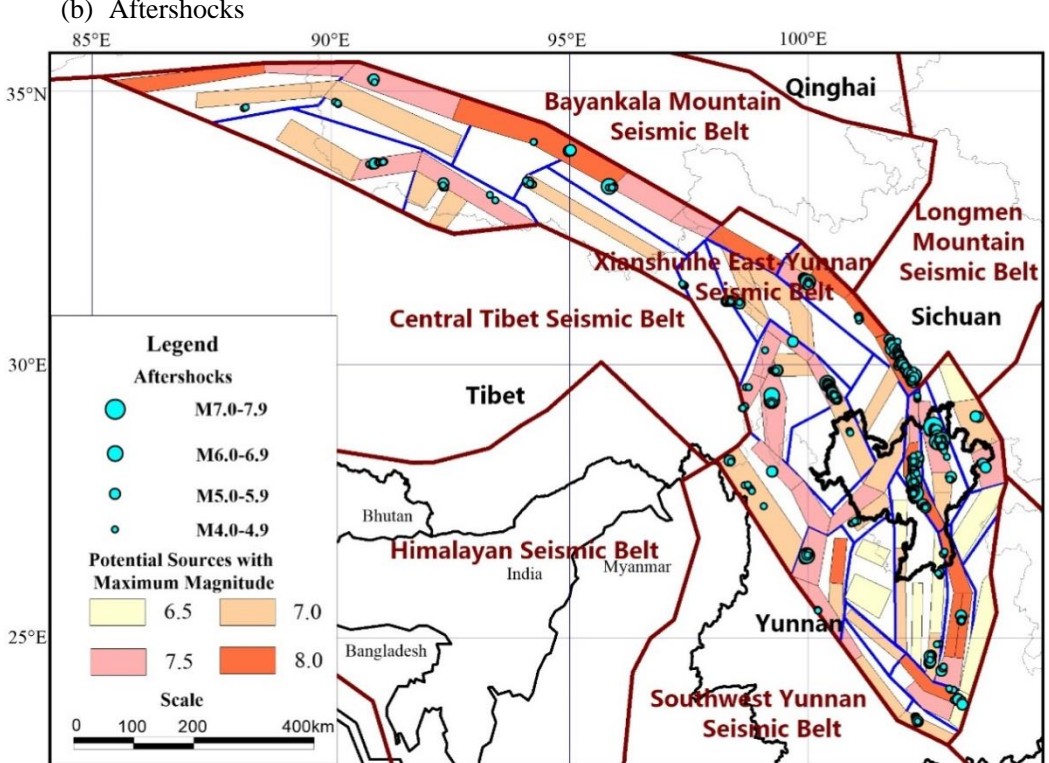

(b) Aftershocks

**Figure 5 Distribution of 1-year mainshocks with M ≥ 6 and their corresponding aftershocks in 100 samplings**

After the aftershocks are obtained, the mainshocks and aftershocks are combined, the ground motion value of the site is calculated by using the ground motion prediction equations (GMPEs), and the exceedance probability for a specific case is counted; thus, a probabilistic seismic hazard analysis that considers aftershocks can be carried out. Figure 6 shows the calculation process for this analysis. According to the Seismic Ground Motion Parameters Zonation Map of China (GB18306-2015), the GMPEs of peak ground acceleration (PGA) suitable for the Xichang area are as follows (Xiao, 2011):

When $M < 6.5$,

$$\begin{cases} \log_{10} \bar{G}_l(M,R) = 2.331 + 0.646M - 2.431\log\left(R + 2.647\exp(0.366M)\right) \\ \log_{10} \bar{G}_s(M,R) = 1.017 + 0.614M - 1.866\log\left(R + 0.612\exp(0.457M)\right) \end{cases}, \quad (13\text{-}1)$$

When $M \geq 6.5$,

$$\begin{cases} \log_{10} \bar{G}_l(M,R) = 3.846 + 0.413M - 2.431\log\left(R + 2.647\exp(0.366M)\right) \\ \log_{10} \bar{G}_s(M,R) = 2.499 + 0.388M - 1.866\log\left(R + 0.612\exp(0.457M)\right) \end{cases}, \quad (13\text{-}2)$$

where $G(M,R)$ is the peak ground acceleration (PGA), $M$ is the magnitude, $R$ is the epicentral distance, and the other coefficients are obtained by regression.

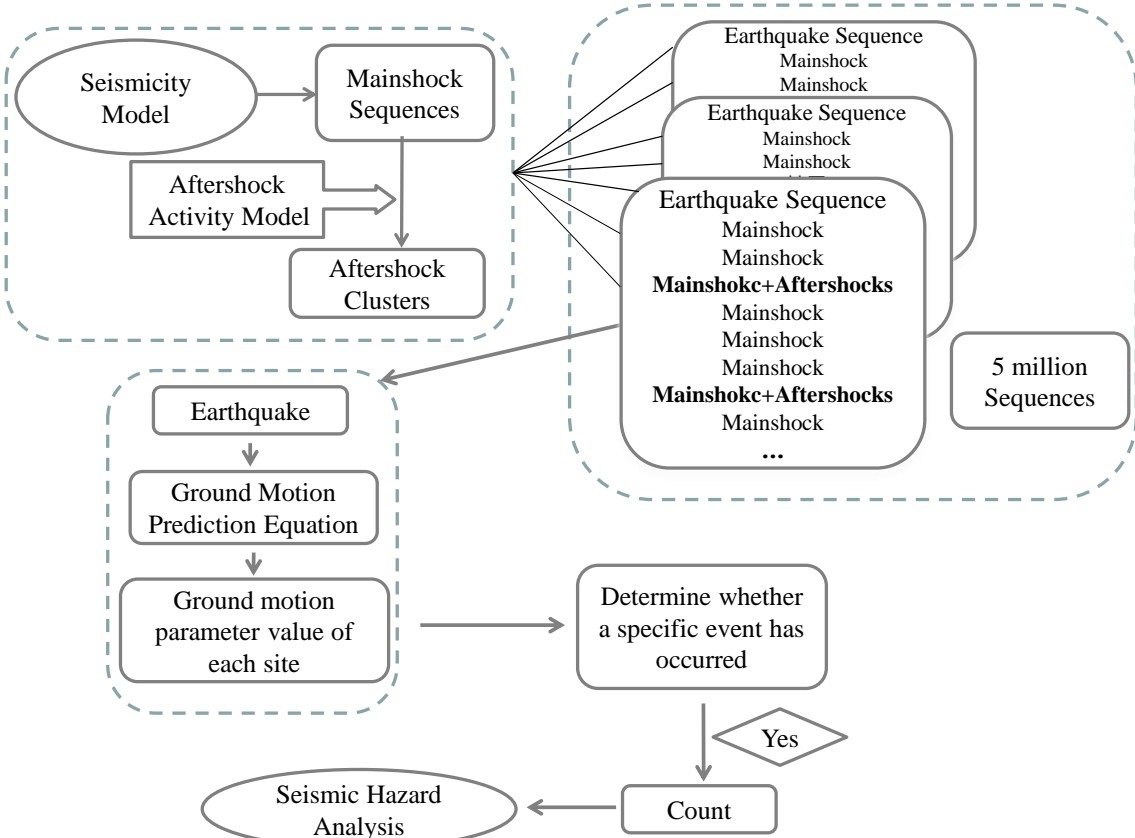

**Figure 6 Flow chart of seismic hazard analysis when considering aftershocks**

**4 Influence of aftershocks on probabilistic seismic hazard analysis**

To calculate the impact of aftershocks on seismic hazard analysis, Xichang and its surrounding areas were divided into a $0.1°{\times}0.1°$ grid (see Fig. 3), and the PGA values of the 50-year exceedance probability of 10% and 2% were calculated for each 415 grid point. The results of the calculation with and without aftershocks were compared.

Figure 7 shows the PGA (gal) contour map of the 50-year exceedance probability of 10% in Xichang and its surrounding areas calculated without and with aftershocks as well as the aftershock impact rate distribution map.

To calculate the aftershock impact rate, we take the difference between the 420 calculation results of the aftershock model and the calculation results of the model without aftershocks and divide that value by the calculation results of the model without aftershocks. That is:

$$impact\ rate = \frac{\left(results\ with\ aftershocks\ -\ results\ without\ aftershocks\right)}{results\ without\ aftershocks}$$

The maximum impact rate of aftershocks in Xichang and its surrounding areas is 55%, the minimum is 0%, and the average is 10%. Aftershocks have the largest impact in the Xichang urban area, where there was a M7 earthquake in 814, a M7.5 earthquake in 1536 and a M7.5 earthquake in 1850. The upper limit of magnitude of the potential source area is 8.0.

Figure 8 shows the PGA (gal) contour map of the 50-year exceedance probability of 2% in Xichang and its surrounding areas calculated with and without aftershocks. Additionally, this figure also shows the aftershock impact rate distribution map. The maximum impact rate of aftershocks in Xichang and its surrounding areas is 72%, the minimum is 0%, and the average is 10%. The greatest impact of aftershocks is also in the Xichang urban area, where there was a M7 earthquake in 814, a M7.5 earthquake in 1536 and a M7.5 earthquake in 1850. The upper limit of magnitude of the potential source area is 8.0. In this calculation, only mainshocks with $M \geqslant 6.0$ generate aftershocks. Therefore, the calculated results are consistent with the aftershock model. The seismic hazards for sites with different seismic backgrounds are affected by aftershocks to different degrees.

(a) without aftershocks

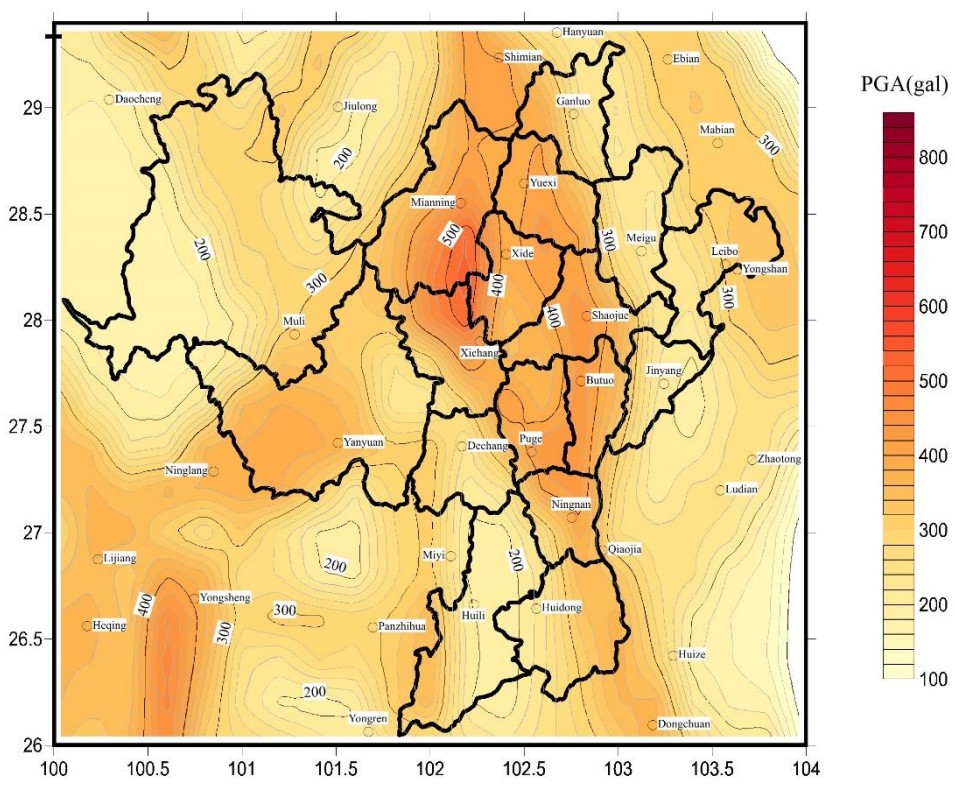

(b) with aftershocks

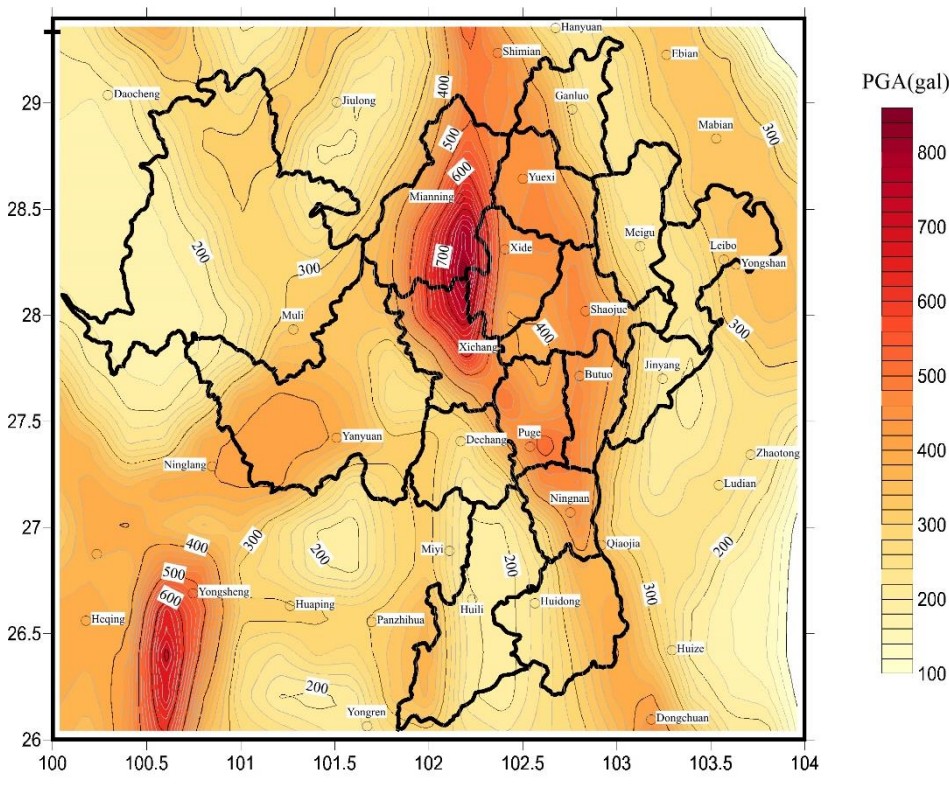

(c) the aftershock impact rate

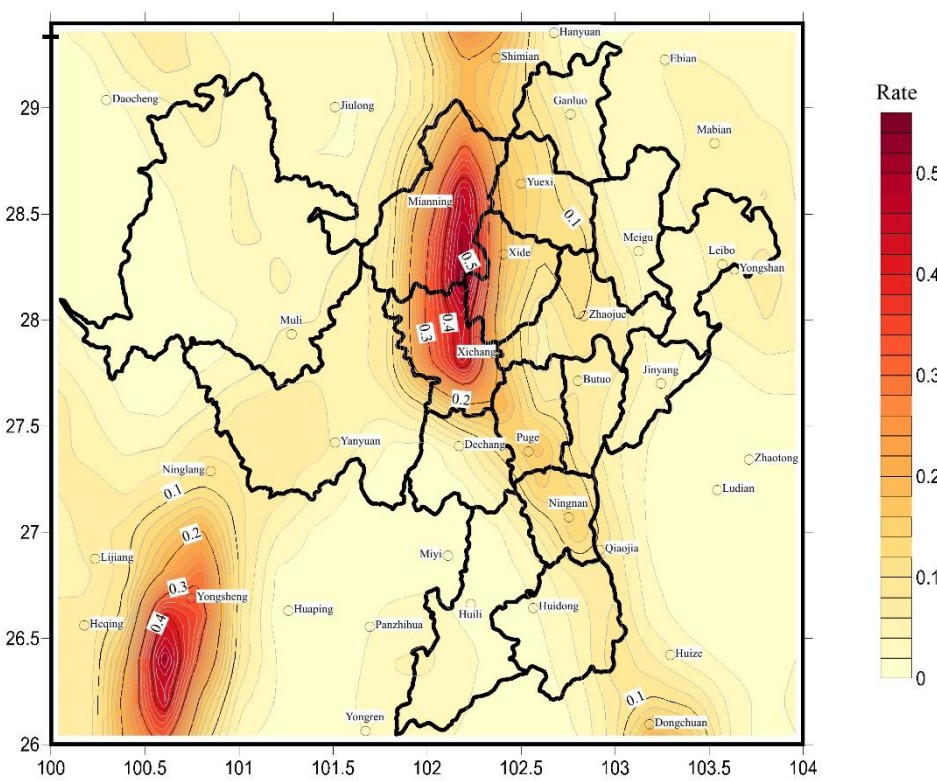

**Figure 7 Comparison of the aftershock impacts on PGA (gal) with a 10% exceedance probability over 50 years in Xichang and its surrounding areas. (a) PGA(gal) contour map of the 50-year exceedance probability of 10% considering only the mainshocks; (b) PGA(gal) contour map of the 50-year exceedance probability of 10% considering the mainshocks and aftershocks simultaneously; (c) distribution map of the aftershock impact rate, aftershock impact rate = (calculation results of model with aftershocks - calculation results of model**

**without aftershocks)/calculation results of model without aftershocks.**

(a) without aftershocks

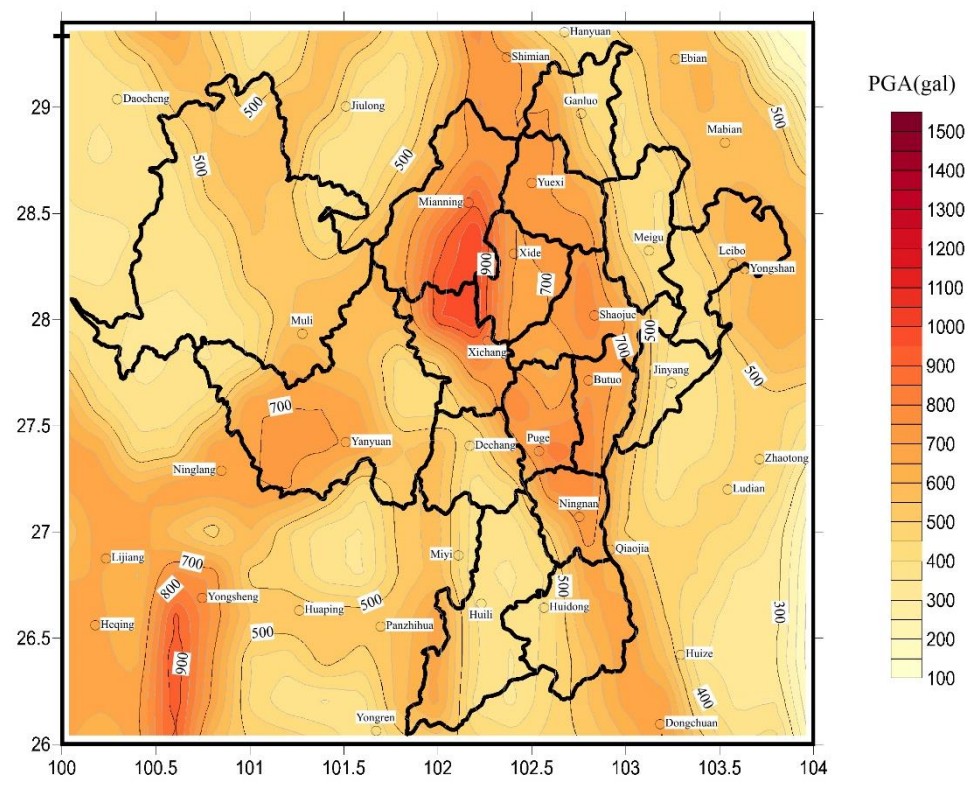

(b) with aftershocks

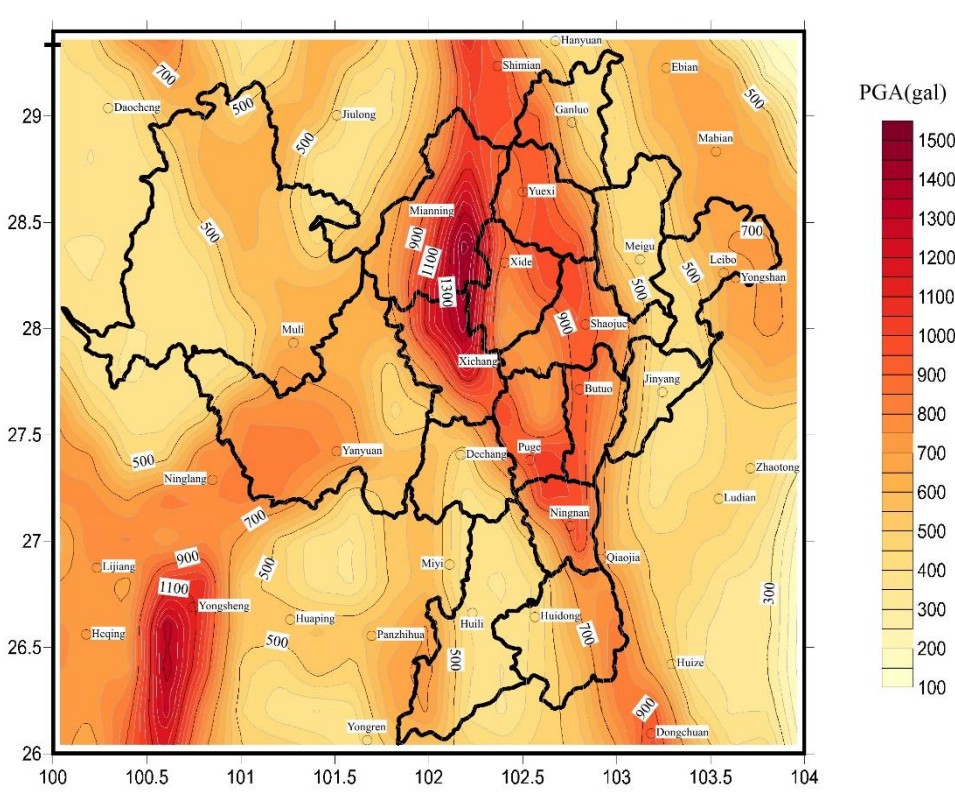

(c) the aftershock impact rate

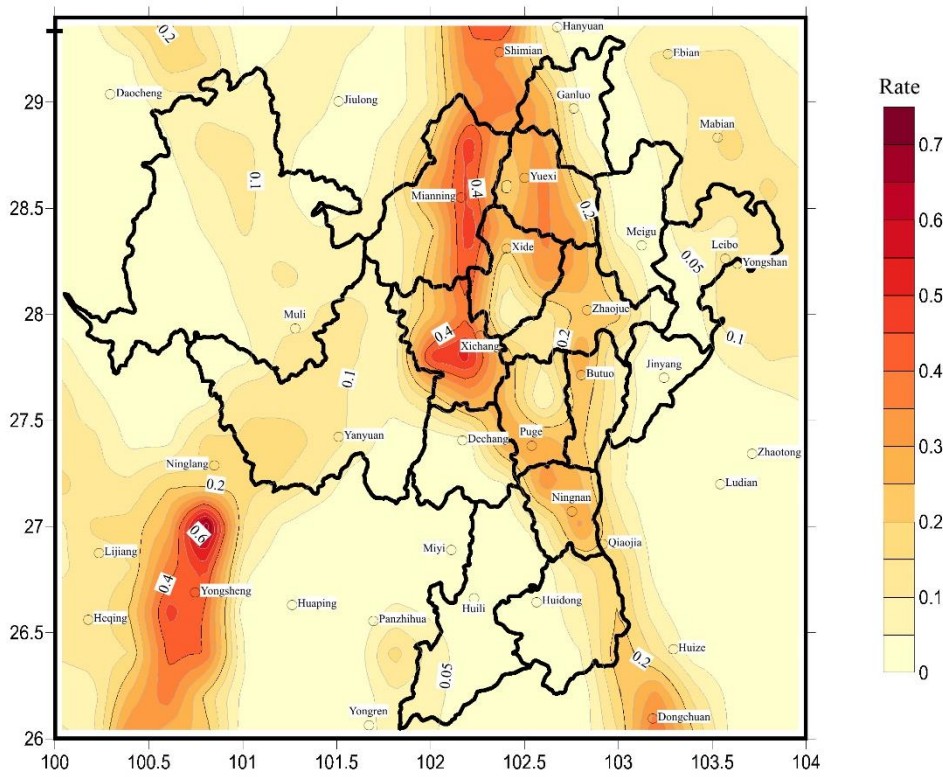

**Figure 8 Comparison of the aftershock impacts on PGA (gal) with a 2% exceedance probability over 50 years in Xichang and its surrounding areas. (a) PGA(gal) contour map of the 50-year exceedance probability of 2% considering only the mainshocks; (b) PGA(gal) contour map of the 50-year exceedance probability of 2% considering the mainshocks and aftershocks simultaneously; (c) distribution map of the aftershock impact rate , aftershock impact rate = (calculation results of model with aftershocks - calculation results of model without aftershocks)/calculation results of model without aftershocks.**

## 5 Comparison with the ETAS model

The ETAS model is a widely used statistical method for capturing short-term spatiotemporal earthquake clustering. However, its application is occasionally impeded by the challenge of estimating a substantial number of unknown parameters. Recent advancements in ETAS formulations introduce spatial and temporal variability in certain parameters, further complicating their estimation process. Mancini and Marzocchi (2023) introduced a simple ETAS method called SimplETAS. The basic idea behind SimplETAS is that the earthquake clustering process in crustal regions is time- and space- independent, a premise substantiated by empirical analyses conducted by Stallone and Marzocchi (2019).

The functions adopted in SimplETAS are defined as follows:

$$\lambda(t,x,y,m) = [\nu\mu(x,y) + \sum_{i:t_i<t} \kappa(m_i)g(t-t_i)f(x-x_i, y-y_i; m_i)]s(m) \quad (14)$$

$$\kappa(m_i)=Ae^{\alpha(M_i-M_{min})} \tag{15}$$

$$g(t-t_i)=\frac{c^{p-1}(p-1)}{(t-t_i+c)^p} \tag{16}$$

$$f(x-x_i,y-y_i|m_i)=\frac{q-1}{\pi De^{\gamma(M_i-M_{min})}}[1+\frac{(x-x_i)^2+(y-y_i)^2}{De^{\gamma(M_i-M_{min})}}]^{-q} \tag{17}$$

in which $A$ is the productivity; $\alpha$ is the coefficient of the exponential magnitude-dependent productivity law; $c$ and $p$ are the time constant and the exponent of the modified Omori law, respectively; $q$ and $D$ define the spatial distribution of triggered events; $\gamma$ accounts for the correlation between the aftershock area and the magnitude of the triggering event; $t_i$, $x_i$, and $y_i$ are the temporal and spatial distances of the $i$th past earthquake from the present time $t$ and from the considered location $(x, y)$, respectively.

In the model, the seven parameters in the conventional ETAS formulation governing earthquake clustering, namely $\{\alpha, p, c, D, \gamma, q, \beta\}$, are predetermined. Only the total background seismicity rate ($v$) and the seismic productivity ($A$) remain to be estimated, which exhibit significant variations depending on the region. The SimplETAS model can work as well as the ETAS model. Therefore, in this study, we follow the SimplETAS model and use their codes to simulate 10,000 sets of earthquake catalogs in Xichang and the surrounding areas for comparison analysis. Referring to Mancini and Marzocchi (2023), Table 4 shows the corresponding parameters. The background seismicity spatial PDF is shown in Figure 9. When estimating the PDF, the M4.0+ earthquake catalog from January 1, 1970, to August 23, 2023 is used. The primary catalog spans from January 1, 1975, to August 23, 2023, while the auxiliary catalog covers the period from 1400 to August 23, 2023. Figure 10 shows the distribution map of the simulated 10,000 earthquake catalogs by SimplETAS.

**Table 4 The SimplETAS parameters used for simulations, with Mmin=3.95, which is similar to Mancini and Marzocchi (2023).**

| Parameter | Value | Type |
| --- | --- | --- |
| V(eqks/yr) | 23.8394 | estimated |
| $\mu$(x,y) | Background seismicity spatial PDF | From Dr. Li |

| | | |
|---|---|---|
| A | 0.0212 | estimated |
| $b$ | 1.0 | fixed |
| $c$ | 0.005 | fixed |
| $p$ | 1.15 | fixed |
| $D(km^2)$ | 1 | fixed |
| $q$ | 1.5 | fixed |
| $\gamma$ | 1.5 | fixed |
| $\alpha, \beta$ | $b \times \ln(10)$ | fixed |

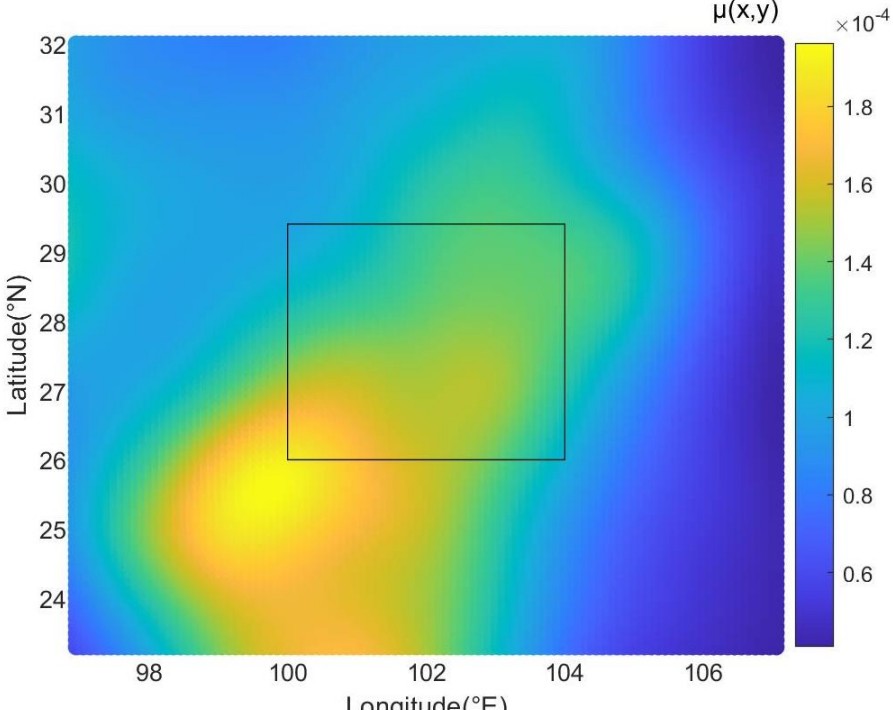

**Figure 9 The estimated background seismicity spatial PDF $\mu$(x,y) for the Sichuan-Yunnan region. The black rectangle represents the area where Xichang is located.**

From Figure 10, it can be seen that there is a significant difference between the location of earthquake clusters simulated by SimplETAS and the distribution of the main- and aftershocks in Section 3.2.

      The potential source models we employed to simulate earthquake catalogs in Section 3 comprehensively consider various data, including paleoearthquakes,
historical earthquakes, seismogenic structures, stress-strain fields. These data help constrain the locations of earthquakes, especially those of high magnitude. However, it's important to note that the ETAS model is an empirical statistical model, relying on

earthquake catalogs as its fundamental data. This distinction makes it challenging to draw direct comparisons between the two models. To address this limitation, it is essential for future research to explore the incorporation of more physics-based models to establish comparative bridges. However, this endeavor goes beyond the scope of the current study.

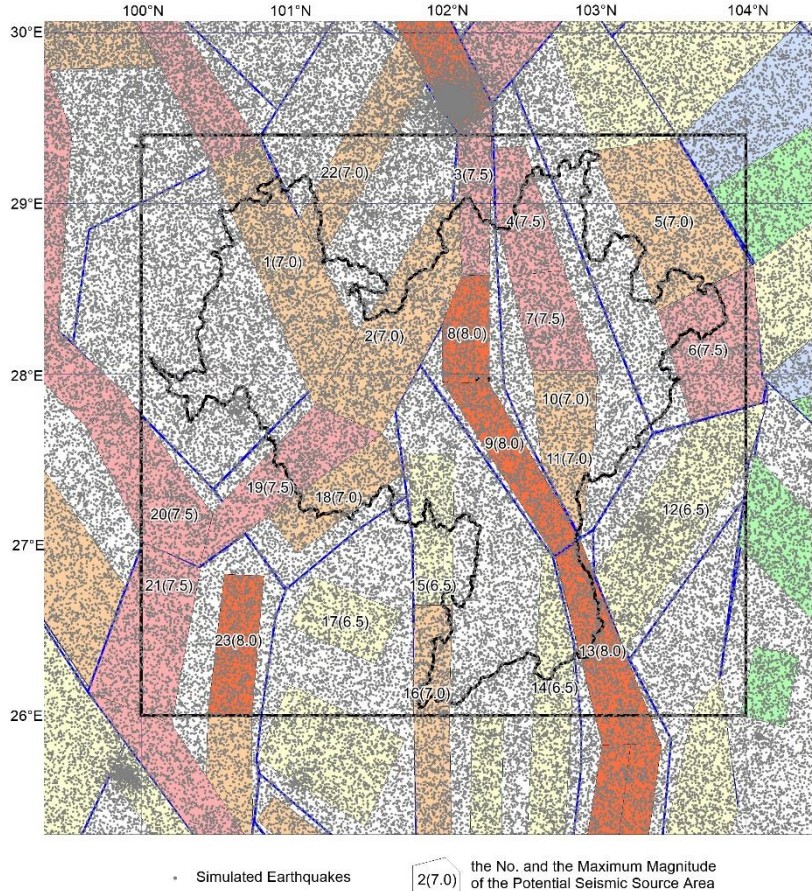

**Figure 10 The distribution map of the simulated 10,000 earthquake catalogs by SimplETAS. The black rectangle represents the area where Xichang is located.**

## 6 Discussion

Several studies have shown that the estimates of the ETAS parameters are highly susceptible to the assumptions made, such as the magnitude cutoff, time dependency of the background rate, anisotropic aftershock triggering, and aftershock incompleteness (Seif et al., 2017; Zhuang et al., 2017). Bearing this in mind, it becomes apparent that comparing parameter values across different studies using diverse catalogs (with variations in quality, magnitude of completeness, and spatial and temporal windows) is not a straightforward task. Moreover, the inherent statistical correlation among the parameters further complicates the comparison process.

The outcomes of the study conducted by Lacoletti et al. (2022) suggest that the traditional region-wide calibration approach is inadequate for constructing an ETAS model suitable for simulation-based PSHA. Generally, sequence-averaged ETAS models prove to be more acceptable, exhibiting both a higher number of aftershocks and consistent spatial and magnitude–frequency distributions. Nevertheless, numerous regions (such as Xichang) face a challenge due to an insufficient occurrence of active sequences within the required (short and recent) period according to the method's criteria.

Šipčić et al. (2022) conducted a comparison of three alternative models (Poisson, Omori, and ETAS) under two different initial conditions: an "unconditional" case, with initial conditions characterized by average seismicity, and a "conditional case," incorporating initial conditions of an ongoing active earthquake sequence. As expected, the traditional Poissonian approach for earthquake occurrence modeling tends to provide lower hazard estimates. As anticipated, the traditional Poissonian approach for earthquake occurrence modeling tends to yield lower hazard estimates. The inclusion of aftershocks in the Omori model and consideration of all events in the ETAS model significantly enhance hazard estimates, providing more realistic values by not solely accounting for the effect of the largest events, as seen in the case of the Poissonian approach.

In our study, we have examined the classical Poissonian model that considers only mainshocks and the model that combines the Poissonian model for mainshocks and the Omi-R-J model for aftershocks, which is considered an approach for clustered seismicity modeling that is less complicated than ETAS, and the Omi-R-J model is sensitive to the identification of mainshocks.

The significant feature of our study is the simulation of the mainshocks based on the potential source model and the seismicity model of the Seismic Ground Motion Parameter Zoning Map of China (GB18306-2015). These models comprehensively consider various data, such as paleoearthquakes, historical earthquakes, seismogenic structures, stress-strain fields, and provide probability functions for the spatial distribution of earthquakes with different magnitude ranges (Gao, 2015), thereby limiting the location of mainshocks (especially high magnitude earthquakes). After the

determination of the mainshocks, the aftershocks are distributed around the mainshocks. However, the ETAS model is an empirical statistical model, and the fundamental data are only earthquake catalogs. Therefore, the accuracy of the ETAS model depends on having well-characterized catalogs. These findings suggest the need to additionally investigate and improve the models through more sophisticated statistics and physics-based models (Hardebeck et al., 2023).

## 7 Conclusions

In this study, a probabilistic seismic hazard analysis based on the Monte Carlo method was combined with the Omi-R-J model to systematically study how aftershocks impact seismic hazard analyses in Xichang city and the surrounding areas. The results show that in areas with moderate to strong seismic backgrounds, the influence of aftershocks on probabilistic seismic hazard analysis can exceed 50%. Aftershocks are typically ignored in traditional probabilistic seismic hazard analyses, which underestimates the seismic hazard to some extent and may cause potential risks. Our results suggest that the impact of aftershocks should be properly considered during future probabilistic seismic hazard analyses, especially in areas with moderate to strong seismic activity backgrounds and in areas prone to secondary disasters such as landslides and mudslides.

The model settings adopted for the calculation processes presented in this study can be modified according to the actual situation and specific requirements. The Monte Carlo method is highly adaptable and can take into account different parameters in different models. In future work, we can attempt to adjust the initial magnitude of the mainshock and the aftershock. Additionally, we can adjust the duration of the aftershock and use different mainshock models and aftershock models to study how aftershocks impact probabilistic seismic hazard analysis.

This work provides a scientific basis for governmental departments to minimize disaster losses and formulate corresponding earthquake prevention and disaster mitigation measures. Furthermore, this work plays very important roles in engineering decision making and judgment, the implementation of catastrophe insurance, and other fields.

## Data availability

All raw data can be provided by the corresponding authors upon request.

## Author contributions

All authors contributed to the study conception and design. Material preparation, data collection and analysis were performed by Qing Wu and Guijuan Lai. The first draft of the manuscript was written by Qing Wu and all authors commented on previous versions of the manuscript. All the authors have read and approved the final manuscript.

## Competing interests

The authors declare that they have no conflict of interest.

## Acknowledgments

We thank the Compiling Committee of the Seismic Ground Motion Parameters Zonation Map of China for providing the seismic source zone model, seismicity model and GMPE model data. Thanks are extended to Dr. Zongchao Li for his help in drawing the map showing the seismic event distribution and tectonic background in Xichang and its surrounding areas. Thanks are also extended to Dr. Jiawei Li for his assistance with the ETAS model. The authors thank Matteo Taroni and two anonymous reviewers for the insightful comments, which helped improve the quality of this article.

## Financial support

This work was supported by the National Key Research and Development Program (grant number 2022YFC3003502), the independent project initiated by the Institute of Geophysics, China Earthquake Administration (grant number JY2022Z41) and the National Key Research and Development  Program of Xinjiang Uygur Autonomous Region (grant number 2020B03006-4).

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
