# Peer review of "The influence of aftershocks on seismic hazard analysis: A case study from Xichang and the surrounding areas"

_Natural Hazards and Earth System Sciences, 2023_

## Referee Comment (RC2)

[referee-annotated manuscript omitted]

---

## Referee Comment (RC3)

[referee-annotated manuscript omitted]

---

## Author Comment (AC1)

1. Line 200. How are the probabilities of seismic hazard obtained? I checked Wu et al. 2020, but did not think that the details had been given in it.

Reply: Thanks for your comments. The probabilities of seismic hazard are obtained based on Monte Carlo method. The following is the detail process of the method

The Monte Carlo method uses random numbers to perform computer simulations. The basic idea is that when the number of experiments is large sufficiently, the frequency of an event appears to approximate the probability of occurrence of the event.

Based on the geophysical data of various regions in China, the seismic zoning map of China (GB18306-2015) shows the seismic zones and potential source areas, has established the corresponding probability model and spatial distribution model of earthquake occurrence, and gives the basic parameters of each seismic zone. Fig. 1 shows the potential seismic sources around Xichang.

[Figure]

**Fig. 1 The calculate sites and the potential seismic sources around Xichang**

According to the basic assumptions and seismicity parameters of the zoning map (Table 1), the following steps are used to synthesize the sets of earthquake sequences (Guo, 2008; Wu & Gao, 2018):

(1)Based on the assumption that the occurrence of earthquakes in seismic zones satisfies Poisson distribution, the time length T of the simulated earthquake sequence and the average annual occurrence rate $v_4$ of earthquakes with magnitude 4 and above in the seismic zone should be determined firstly. Randomly generate a Poisson distribution random number L with T and $v_4$ as parameters, then L is the number of earthquakes in the seismic zone for the length of time T to be simulated.

**Table 1 List of seismicity parameters of potential seismic sources around Xichang**

| No. | $M_{UZ}$ | b value | $v_4$ | Strike | No. | $M_{UZ}$ | b value | $v_4$ | Strike |
|-----|----------|---------|-------|--------|-----|----------|---------|-------|--------|
| 1 | 7.0 | 0.85 | 32 | 120° | 12 | 6.5 | 0.85 | 32 | 50° |
| 2 | 7.0 | 0.85 | 32 | 55° | 13 | 8.0 | 0.85 | 32 | 120° |
| 3 | 7.5 | 0.85 | 32 | 90° | 14 | 6.5 | 0.85 | 32 | 80° |
| 4 | 7.5 | 0.85 | 32 | 115° | 15 | 6.5 | 0.85 | 32 | 90° |
| 5 | 7.0 | 0.85 | 32 | 120° | 16 | 7.0 | 0.85 | 32 | 90° |
| 6 | 7.5 | 0.85 | 32 | 120° | 17 | 6.5 | 0.85 | 32 | 150° |
| 7 | 7.5 | 0.85 | 32 | 115° | 18 | 7.0 | 0.85 | 32 | 30° |
| 8 | 8.0 | 0.85 | 32 | 90° | 19 | 7.5 | 0.85 | 32 | 45° |
| 9 | 8.0 | 0.85 | 32 | 125° | 20 | 7.5 | 0.85 | 32 | 120° |
| 10 | 7.0 | 0.85 | 32 | 90° | 21 | 7.5 | 0.85 | 32 | 55° |
| 11 | 7.0 | 0.85 | 32 | 80° | 22 | 7.0 | 0.85 | 32 | 55° |

(2) Based on the assumption that the magnitude distribution of seismic zones satisfies the truncated Gutenberg-Richter relationship (magnitude-frequency relationship), and the minimum magnitude level $M_0$ and the maximum magnitude $M_{UZ}$, the magnitude of earthquakes to be simulated are determined.

The magnitude-frequency relationship is represented as:

$$\log N = a - bM \tag{1}$$

Where $a$ and $b$ are coefficients, $N$ is the number of earthquakes whose magnitude is equal to or greater than $M$, and the initial magnitude of the zoning map is 4. The cumulative number of earthquake events is:

$$N(M) = e^{a-bM} \tag{2}$$

If take $\Delta M = 0.1$, then

$$N(M) > N(M + \Delta M) \tag{3}$$

Take M=4.1, 4.2, 4.3,…, $M_{UZ}$ .Generate a random number $u$ satisfied uniform distribution between 0 and 1. Determine whether

$$u \in \frac{N(\text{M}+\Delta\text{M})}{N(4)} \sim \frac{N(\text{M})}{N(4)} \tag{4}$$

If the above formula is true, the magnitude $M$ of an earthquake event is determined.

(3) Determination of epicenter location. Firstly, the potential source area $H$ where the earthquake located should be determined. According to the magnitude $M$ determined in the previous step, the magnitude range $d$ which the earthquake belongs to is determined. Because the probability $P_d(\text{h})$ of each magnitude range locating in each potential source area is known, then generate a random number $u$ satisfied uniform distribution between 0 and 1. Determine whether

$$u \in \sum_{h=1}^{H-1} P_d(\text{h}) \sim \sum_{h=1}^{H} P_d(\text{h}) \tag{5}$$

If so, the potential source area $H$ where the earthquake event is located is determined. Based on the assumption that the epicenter is evenly distributed in the potential source area, a point is randomly selected in the potential source area $H$ as the epicenter location of an earthquake.

(4) According to the azimuth of the potential source area, the azimuth of the earthquake is determined.

So far, the basic elements of an earthquake have been determined. Repeat (2) ~ (4) steps until the required number $L$ of earthquakes in the seismic zone, taking into account all possible seismic zones that may affect the site, thus determining a seismic sequence and completing one sampling.

If the time length $T$ is set to one year, the seismic sequence obtained by one sampling is called one-year seismic sequence in this paper. The time length is set to 10 years, which is called 10-year earthquake sequence.

According to the principle of Monte Carlo method, the more samplings, the more stable the result is. But the more samplings, the more calculations. Therefore, in order to consider the accuracy of the results and the calculation quantity as a whole, it is necessary to carry out experiments with different number of samplings. When the calculation results tend to be stable, it is considered that there is no need to increase the number of samplings.

For each earthquake in seismic sequences, the peak ground acceleration (PGA) of each site is calculated by the optimal ellipse search algorithm through the ground motion prediction equations (GMPEs).

For 5000000 simulations of a 1-year earthquake sequence, if a site is affected by ground motions exceeding specific values, the sequence is identified as 1. The sum of earthquake sequences identified as 1 is counted, and is divided by the total number of earthquake sequence simulations of 5000000, that is the annual exceedance probability of specific ground motions. Through the annual exceedacne probability, the 50-year exceedance probability of 10% and 2% can be calculated.

That is how the probabilities of seismic hazard are obtained.

We will add the details into the manuscript during the revision process.

**2. How are the aftershocks simulated?**

Reply: We used the Omi-R-J model to calculate the aftershock sequence parameters of 4 M7.0+ and 40 M4.5-7.0 mainshocks, which occurred in the study region (the Xianshuihe East-Yunnan seismic belt) from 1970 through 2018. The results were shown in Table 2 in lines 230-255, we introduced that the median values of the estimated p, c, K and b from the 44 aftershock sequence samples are 0.8747, 0.0187, 0.0133 and 0.8361, respectively. When the magnitude threshold for the mainshock is met (M ≥ 6.0 in this study), the aftershocks are simulated as follows:

(1) With respect to the magnitude and time: the minimum magnitude of the aftershock sequence is set to 4.0, and the maximum magnitude is equal to the magnitude of the mainshock. The aftershock sequence satisfies the magnitude-frequency relationship $N(M){=}10^{a-bM}$. The aftershock occurrence time $t$ is within 30 days after the mainshock and follows the Omori-Utsu formula $N(t){=}\dfrac{K}{(t+c)^{p}}$. According to the median value of p, c, K and b, the magnitude and time series of aftershocks with M ≥ 4 are simulated.

(2) With respect to the spatial distribution: according to the empirical relationship between the magnitude of the mainshock and the rupture scale (Wells & Coppersmith, 1994), the rupture length L and width W are calculated by:

$$L{=}10^{(-3.22+0.69M\text{w})} \qquad (6)$$

$$W{=}10^{(-1.01+0.32M\text{w})} \qquad (7)$$

The rupture strike is taken the same as the direction of the mainshock, and the model of Felzer & Brodsky (2006) is adopted; that is, the aftershock density decays exponentially with increasing distance r from the fault, $\rho(r) = cr^{-n}$, where $n$ is 1.37, and $c$ is a constant. Thus, the locations of the aftershock epicenters can be determined.

(3) The number of aftershocks. We have accounted the number of M4.0+ aftershocks for M5.0+ mainshocks in the Chinese mainland and its surrounding area, and found that when the mainshock is greater than 6.0, the number of M4.0+ aftershocks within a month (30 days) increase with the magnitude of mainshock, and they meets the statistical relationship: $\log_{10} (N)=0.84M-4.57$ (shown by the red line in the middle in Fig. 2, which fluctuates within the range of ±0.8 (shown by the other two red lines), and obeys the Normal distribution under the linear coordinates. The number of aftershocks corresponding to a certain magnitude is generated according to this law.

[Figure]

**Fig. 2 The M5.0+ mainshocks and the number of their M4.0+ aftershocks for the Chinese mainland and its surrounding area**

3. How is the ETAS model, given in Section 2.2, used in the evaluation?

Reply: When using the ETAS model to fit the parameters of aftershock sequences, there is a high requirement for the number of aftershocks with magnitude higher than the completeness magnitude, and only small part of earthquake sequence samples can meet the conditions. Therefore, we used the ETAS model to calculate the aftershock

sequence parameters for 4 M7.0+ mainshocks. For moderate size of mainshocks, the aftershock sequences are usually not complex. We choose to use the Omi-R-J model to calculate the parameters of their aftershock sequences. Compared to the R-J model, a detection rate function is introduced to describe the detection rate of the incomplete part of the earthquake catalog, which considers aftershocks below the completeness magnitude in the early stage of the earthquake sequence during the model parameter fitting. The ETAS model was used for comparison. We will delete it and focus on the Omi-R-J model in the revision process.

Other:

1. Line 150. Is EM algorithm necessary? How are p, c, k estimated?

Reply:
(1) Yes, we think it is necessary. In the early period after a mainshock, the waveforms of small earthquakes were submerged by the waveforms of large earthquakes, making it difficult to identify small earthquakes and resulting in the lack of catalog of small earthquakes. The EM algorithm is based on the super parameter estimation of the Newton iterative algorithm. It can optimize the parameters in case of missing small earthquakes in the early period, reducing the error of the Newton iterative algorithm and obtain more objective parameters.
(2) The earthquake detection rate function considering incomplete earthquake records can be expressed as $v(t,M) = \lambda(t,M)q(M|\mu(t),\sigma)$. The logarithmic likelihood function related to parameters p, c, k is:

$$\ln L(k,c,p) = \sum_{M_i \geq M_c} \ln v(t,M) - \int_{M_c}^{\infty} dM \int_0^T dt v(t,M) \tag{8}$$

Where $t_i \ and \ M_i$ are the time and magnitude of the i-th aftershock that occurred within the "learning period" [0, T] during model fitting.

We will add the details into the manuscript during the revision process.

---

## Author Response (AR1)

Reviewer #1

1. Line 200. How are the probabilities of seismic hazard obtained? I checked Wu et al. 2020, but did not think that the details had been given in it.

Reply: Thanks for your comments. The probabilities of seismic hazard are obtained based on Monte Carlo method. The following is the detail process of the method:

Based on the geophysical data of various regions in China, the seismic zoning map of China (GB18306-2015) shows the seismic zones and potential source areas, has established the corresponding probability model and spatial distribution model of earthquake occurrence, and gives the basic parameters of each seismic zone. Fig. 3 in the revised manuscript shows the potential seismic sources in and around Xichang.

According to the basic assumptions and seismicity parameters of the zoning map (Table 3 of the revised manuscript), the following steps are used to synthesize the sets of earthquake sequences (Wu & Gao, 2018, Wu et al., 2020):

(1) Based on the assumption that the occurrence of earthquakes in seismic zones satisfies the Poisson distribution, the time length T of the simulated earthquake sequence and the average annual occurrence rate $v_4$ of earthquakes with magnitude 4 and above in the seismic zone should be determined first. Then, a Poisson distribution random number $L$ is generated with T and $v_4$ as parameters, where $L$ is the number of earthquakes in the seismic zone for the length of time T to be simulated.

(2) Based on the assumption that the magnitude distribution of seismic zones satisfies the truncated Gutenberg-Richter relationship (magnitude-frequency relationship), with the minimum magnitude level $M_0$ and the maximum magnitude level $M_{UZ}$, the magnitudes of earthquakes to be simulated are determined.

The magnitude-frequency relationship is represented as:

$$\log N = a - bM \tag{1}$$

where $a$ and $b$ are coefficients, $N$ is the number of earthquakes whose magnitude are equal to or greater than $M$, and the initial magnitude of the zoning map is 4. The cumulative number of earthquake events is:

$$N(M) = e^{a-bM} \tag{2}$$

If we take $\Delta M = 0.1$, then

$$N(M) > N(M + \Delta M) \tag{3}$$

Based on $M$=4.1, 4.2, 4.3,…, $M_{UZ}$, a random number $u$ that satisfies a uniform distribution between 0 and 1 is generated. Then, the following is determined:

$$u \in \frac{N(M + \Delta M)}{N(4)} \sim \frac{N(M)}{N(4)} \tag{4}$$

If the above formula is true, the magnitude $M$ of an earthquake event is determined.

(3) Determination of epicenter location. First, the potential source area $H$ where the earthquake is located should be determined. According to the magnitude $M$ determined in the previous step, the magnitude range $d$ to which the earthquake belongs is determined. Because the probability $P_d(\text{h})$ of each magnitude range in each potential source area is known, a random number $u$ is generated that satisfies a uniform distribution between 0 and 1. The following is then determined:

$$u \in \sum_{h=1}^{H-1} P_d(h) \sim \sum_{h=1}^{H} P_d(h) \tag{5}$$

If so, the potential source area $H$ where the earthquake event is located is determined. Based on the assumption that the epicenter is evenly distributed in the potential source area, a point is randomly selected in the potential source area $H$ as the epicenter location of an earthquake.

(4) According to the azimuth of the potential source area, the azimuth of the earthquake is determined.

At this point in the calculation, the basic elements of an earthquake have been determined. Steps (2) ~ (4) are repeated until the required number $L$ of earthquakes in the seismic zone is obtained, accounting for all possible seismic zones that may affect the site, thus determining a seismic sequence and completing one sampling.

If the time length $T$ is set to one year, the seismic sequence obtained by one sampling is called a one-year seismic sequence in this paper. When the time length is set to 10 years, the sequence is called a 10-year earthquake sequence.

For each earthquake in seismic sequences, the peak ground acceleration (PGA) of each site is calculated by the optimal ellipse search algorithm through ground motion

prediction equations (GMPEs).

For 5000000 simulations of a 1-year earthquake sequence, if a site is affected by ground motions exceeding specific values, the sequence is assigned a value of 1. The sum of earthquake sequences identified as 1 is counted, and is divided by the total number of earthquake sequence simulations (i.e., 5000000), resulting in the annual exceedance probability of specific ground motions. Through the annual exceedance probability, the 50-year exceedance probabilities of 10% and 2% can be calculated.

This is how the probabilities of seismic hazard are obtained.

We have added the details into the revised manuscript (Section 3.1 of the revised manuscript, Line 250-Line 310).

2. How are the aftershocks simulated?

Reply: We used the Omi-R-J model to calculate the aftershock sequence parameters of 4 M7.0+ and 40 M4.5-7.0 mainshocks, which occurred in the study region (the Xianshuihe East-Yunnan seismic belt) from 1970 through 2018. The results were shown in Table 2 in the revised manuscript, we introduced that the median values of the estimated *p, c, K* and *b* from the 44 aftershock sequence samples are 0.8747, 0.0187, 0.0133 and 0.8361, respectively. When the magnitude threshold for the mainshock is met (i.g., M ≥ 6.0 in this study), we consider the effect of the aftershocks. According to the Bath's law (Bath,1965), for a M6.0 mainshock, the biggest aftershock is usually 1.2 less than it, which is less than M4.8; As is shown in Fig.4 of the revised manuscript, the number of M4.0+ aftershocks for M6.0-mainshock is mainly less than five. Considering a potential sufficiently large impact on the sites, we take the magnitude threshold as 6.0, and the value can be adjusted as needed.

The aftershocks are simulated as follows:

(1) The minimum magnitude of the aftershock sequence is set to 4.0, and the maximum magnitude is equal to the magnitude of the mainshock. In fact, the magnitude of aftershocks can be greater than that of the mainshock. In this study, we focus on the 'aftershocks', so we adopted the assumption of Iervolino et al. (2014). That method assumes foreshocks do not contribute exceedances, aftershocks do not trigger their own aftershocks, and aftershocks are smaller than

the mainshocks. The aftershock sequence satisfies the magnitude-frequency relationship $N(M)=10^{a-bM}$. The aftershock occurrence time $t$ is within 30 days after the mainshock and follows the Omori-Utsu formula $N(t)=\dfrac{K}{(t+c)^p}$. The time interval between a strong aftershock and the mainshock varies from a few seconds to several years, but most strong aftershocks occur a few days or even a day after the main shock (Japan Meteorological Agency, 2009; Tahir et al., 2012). A length of 30 days is taken as the duration for a simplified calculation, and can be changed as needed. According to the median values of $p$, $c$, $K$ and $b$ and the upper limit of magnitude of the potential sources, the magnitude and time series of aftershocks with M $\geqslant$ 4 are simulated.

(2) According to the empirical relationship between the magnitude of the mainshock and the rupture scale (Wells & Coppersmith, 1994), the rupture length and width are calculated by:

$$L=10^{(-3.22+0.69M_{\mathrm{W}})} \tag{6}$$

$$W=10^{(-1.01+0.32M_{\mathrm{W}})} \tag{7}$$

The rupture strike is taken as the direction of the potential source area where the mainshock is located, and the model of Felzer & Brodsky (2006) is adopted; that is, the aftershock density decays exponentially with increasing distance $r$ from the fault, $\rho(r) = cr^{-n}$, where $n$ is 1.37, and $c$ is a constant. Thus, the locations of the aftershock epicenters can be determined.

(3) The number of aftershocks. We have accounted for the number of M4.0+ aftershocks for M5.0+ mainshocks in the Chinese mainland and its surrounding area, and found that when the mainshock is greater than 6.0, the number of M4.0+ aftershocks within a month (30 days) increases with the magnitude of the mainshock, yielding the statistical relationship: log10 $(N)=0.84M-4.57$ (shown by the red line in Fig. 4 of the revised manuscript). This relationship fluctuates within the range of $\pm0.8$ (shown by the two dotted red lines), and obeys the normal distribution under linear coordinates. The number of aftershocks corresponding to a certain magnitude is generated according to this relationship.

The details can be found in the revised manuscript (Section 3.2 of the revised

manuscript, Line 312-Line 365).

3. How is the ETAS model, given in Section 2.2, used in the evaluation?

Reply: When using the ETAS model to fit the parameters of aftershock sequences, there is a high requirement for the number of aftershocks with magnitude higher than the completeness magnitude, and only small part of earthquake sequence samples can meet the conditions. Therefore, we used the ETAS model to calculate the aftershock sequence parameters for 4 M7.0+ mainshocks. For moderate size of mainshocks, the aftershock sequences are usually not complex. We choose to use the Omi-R-J model to calculate the parameters of their aftershock sequences. Compared to the R-J model, a detection rate function is introduced to describe the detection rate of the incomplete part of the earthquake catalog, which considers aftershocks below the completeness magnitude in the early stage of the earthquake sequence during the model parameter fitting. The ETAS model was used for comparison. We have adopted the SimplETAS model (Mancini and Marzocchi, 2023) to compare with the Omi-R-J results in the discussion section of the revised manuscript (Section 5 of the revised manuscript, Line 471-Line 563).

Other:

1. Line 150. Is EM algorithm necessary? How are p, c, k estimated?

Reply:
(1) Yes, we think it is necessary. In the early period after a mainshock, the waveforms of small earthquakes were submerged by the waveforms of large earthquakes, making it difficult to identify small earthquakes and resulting in the lack of a catalog of small earthquakes. The EM algorithm is based on the super parameter estimation of the Newton iterative algorithm. It can optimize the parameters in the case of missing small earthquakes in the early period, reducing the error of the Newton iterative algorithm and obtain more objective parameters (Section 2.1 of the revised manuscript, Line 154-Line 161).
(2) The earthquake detection rate function considering incomplete earthquake records can be expressed as $\nu(t, M) = \lambda(t, M)q(M|\mu(t), \sigma)$. The logarithmic likelihood function related to parameters $p$, $c$, and $k$ is:

$$\ln L(k, c, p) = \sum_{M_i \geq M_c} \ln \nu(t, M) - \int_{M_c}^{\infty} dM \int_0^T dt \nu(t, M) \tag{8}$$

where $t_i \ and \ M_i$ are the time and magnitude of the $i$-th aftershock that occurred within the "learning period" [0, T] during model fitting.

We have added the details into the revised manuscript (Section 2.1 of the revised manuscript, Line161-Line 169).

Reviewer #2

1.  Line 45, I think that the underestimation is for all areas.

Reply: Thanks for your comment. For the areas with low seismicity or sparse population, the impacts of aftershocks may be very low and even can be ignored.

2.  Line 65, in this list, you missed two important papers, please take a look and describe the results of these two works.

Reply: Thanks for your reminder. In the updated manuscript, we added that "Field et al. (2021) use the Third Uniform California Earthquake Rupture Forecast (UCERF3) ETAS model (UCERF3-ETAS) to evaluate the effects of declustering and Poisson assumptions on seismic hazard estimates. Wang et al. (2022) compared the ETAS-simulated hazard with approximations based on the declustered Poisson approach (DP), the nondeclustered Poisson approach (NDP), and the recently proposed sequence-based PSHA (Iervolino et al., 2014)." Please see the lines 65-70 in the revised manuscript.

3.  Line 65-66, this statement is unclear, you can remove it.

Reply: Thanks for your comment. We have removed that statement.

4.  Line 118-119, this part must be reformulated. An earthquake with magnitude 8.5 for sure will generate more aftershocks of an earthquake with magnitude 3.5; therefore, you have to better explain your point here.

Reply: You are right. The aftershock productivity is generally increase with magnitude. We intend to express that for the same magnitude, the aftershock productivity can vary significantly, but made an incorrect expression. We revised it as

"The parameter $k$ controls the overall aftershock productivities" and deleted the second half of the sentence. Please see the lines 122-123 in the revised manuscript.

Reply: Thanks for your reminder. The unit of the peak ground acceleration (PGA) is gal, and we have defined it in the revision. Please see the lines 419-421 in the revised manuscript.

6. Line 302-307, please better explain what are you talking about. Define the quantities and the unit of measure.

Reply: Thanks for the comment, the aftershock impact rate = (calculation results of model with aftershocks - calculation results of model without aftershocks)/calculation results of model without aftershocks, that is

$$impact\ rate = \frac{(results\ with\ aftershocks\ -\ results\ without\ aftershocks)}{results\ without\ aftershocks}$$

The rate is a percentage. Please see the lines 422-426 in the revised manuscript.

7. Figure 5-6, please define what are you showing in panel a and b, and the relative unit of measure.

Reply: Thanks for the comment. We have revised the titles of the figures, and Figures 5 and 6 correspond to Figures 7 and 8 in the revised version. Figure 7 (a) PGA gal) contour map of the 50-year exceedance probability of 10% considering only the mainshocks; (b) PGA(gal) contour map of the 50-year exceedance probability of 10% considering the mainshocks and aftershocks simultaneously; (c) distribution map of the aftershock impact rate, aftershock impact rate = (calculation results of model with aftershocks - calculation results of model without aftershocks)/calculation results of model without aftershocks. Figure 8 (a) PGA(gal) contour map of the 50-year exceedance probability of 2% considering only the mainshocks; (b) PGA(gal) contour map of the 50-year exceedance probability of 2% considering the mainshocks and aftershocks simultaneously; (c) distribution map of the aftershock impact rate, aftershock impact rate = (calculation results of model with aftershocks - calculation

results of model without aftershocks)/calculation results of model without aftershocks. The rate is a percentage.

8.  Discussion and conclusions, this part is too short and poor. You have to properly discuss the assumption and limitations of your method, and then compare your method with other approaches.

Reply: Thanks for your suggestion. We have reorganized the Discussion section, discussed the assumption and limitations of our method, have adopted the SimplETAS model (Mancini and Marzocchi, 2023) to compare with the Omi-R-J results (Section 5 of the revised manuscript, Line 471-Line 563).

Reviewer #3

1.  Line 59-61, I think you could have done the same in this work and use the result as benchmark to validate your Omi-R-J results.

Reply:Thanks for your suggestion. We have adopted the SimplETAS model (Mancini and Marzocchi, 2023) to compare with the Omi-R-J results in the discussion section of the revised manuscript (Section 5 of the revised manuscript, Line 471-Line 563).

2.  Line 68-72, probably all this procedure would be well (an in a straightforward way) replaceable by ETAS simulations. If you decide not to employ ETAS, then you should justify this choice.

Reply:Thanks for your suggestion. We have adopted the SimplETAS model (Mancini and Marzocchi, 2023) to compare with the Omi-R-J results in the discussion section of the revised manuscript (Section 5 of the revised manuscript, Line 471-Line 563).

3.  Figure 1, for the legend, I would suggest the wording 'M7 to M8', etc.

Reply: Thanks for the comment, we have changed the words of the legend. Please see Figure 1 in the revised manuscript.

4.  Line 117-119, I am not sure this is true, as the triggering potential should be magnitude-dependent, please check or better explain.

Reply:Thanks for your reminder. The aftershock productivity is generally increase with magnitude. We intend to express that for the same magnitude, the aftershock productivity can vary significantly, but made an incorrect expression. We revised it as "The parameter k controls the overall aftershock productivities" and deleted the second half of the sentence. Please see the lines 122-123 in the revised manuscript.

5.  Line 124-126, again, ETAS is simple as well and could be implemented here. If there are issue with ETAS related to fitting parameters (and I would agree), maybe there are ways to work them out. In essence, I think the ETAS-oriented approach should be used –at minimum- as a benchmark (i.e., from other publications) to compare your results. For the two points raised above, you might want to check the approaches.

Reply:Thanks for your suggestion. We have adopted the SimplETAS model (Mancini and Marzocchi, 2023) to compare with the Omi-R-J results in the discussion section of the revised manuscript (Section 5 of the revised manuscript, Line 471-Line 563).

6.  Line 243, why? Empirical evidence shows that the magnitude of aftershocks can be larger than the 'mainshock'. Please justify this choice better.

Reply: Thanks for the comment, yes, empirical evidence shows that the magnitude of aftershocks can be larger than the 'mainshock'. In our study, we focus on the 'aftershocks', so we adopted the assumption of Iervolino et al. (2014). That method assumes foreshocks do not contribute exceedances, aftershocks do not trigger their own aftershocks, and aftershocks are smaller than the mainshocks. Please see the lines 330-335 in the revised manuscript.

7.  Line 245, similar to the previous comment, this needs to be better justified.

Reply:The time interval between a strong aftershock and the main shock varies from a few seconds to several years, but most strong aftershocks occur a few days or even a day after the main shock (Japan Meteorological Agency, 2009; Tahir et al., 2012). This project started on a collaboration with an insurance company, who concerned about strong aftershocks within a month, so we have chosen a 30-day timeframe. Take 30 days as the duration is a simplified calculation, and can be changed as needed. Please see the lines 337-342 in the revised manuscript.

8. Line 253-255, this somehow corroborates my suggestions: why not simply simulating a space-time ETAS model?

Reply:Thanks for your suggestion. We have adopted the SimplETAS model (Mancini and Marzocchi, 2023) to compare with the Omi-R-J results in the discussion section of the revised manuscript (Section 5 of the revised manuscript, Line 471-Line 563).

9. Line 318-319, also this choice needs a more thorough justification.

Reply:Thanks for your suggestion. According to the Bath's law (Bath,1965), for a M6.0 mainshock, the biggest aftershock is usually 1.2 less than it, which is less than M4.8; As is shown in Fig.4 of the revised manuscript, the number of M4.0+ aftershocks for M6.0- mainshock is mainly less than five. Considering a potential sufficiently large impact on the sites, we take the magnitude threshold as 6.0, and the value can be adjusted as needed. Please see the lines 325-328 in the revised manuscript.

10. Figure 5-6, you need to give a title to the colorbar (i.e., what it represents) and express its unit of measure.

Reply: Thanks for your suggestion. We have added a title to the colorbar in the updated version, and Figures 5 and 6 correspond to Figures 7 and 8 in the revised version. Figure 7 (a) PGA(gal) contour map of the 50-year exceedance probability of 10% considering only the mainshocks; (b) PGA(gal) contour map of the 50-year exceedance probability of 10% considering the mainshocks and aftershocks

simultaneously; (c) distribution map of the aftershock impact rate, aftershock impact rate = (calculation results of model with aftershocks - calculation results of model without aftershocks)/calculation results of model without aftershocks. Figure 8 (a) PGA(gal) contour map of the 50-year exceedance probability of 2% considering only the mainshocks; (b) PGA(gal) contour map of the 50-year exceedance probability of 2% considering the mainshocks and aftershocks simultaneously; (c) distribution map of the aftershock impact rate, aftershock impact rate = (calculation results of model with aftershocks - calculation results of model without aftershocks)/calculation results of model without aftershocks. The rate is a percentage.

11. Discussion and conclusions, I believe here you have quite a lot to expand. This section currently looks to short.

Reply: Thanks for your suggestion. We have reorganized the Discussion section, discussed the assumption and limitations of our method, and adopted the SimplETAS model (Mancini and Marzocchi, 2023) to compare with the Omi-R-J results in the revised manuscript (Section 5 of the revised manuscript, Line 471-Line 563).

---

## Referee Report (RR1)

**Major points:**

1) I would suggest moving the ETAS simulations part from the "Discussion" section to the earlier sections (maybe 3.2?) where you present the results. It looks strange to see additional modelling introduced and presented in a section where results should only be commented.

2) I find the idea of using the "simplETAS" model appropriate. However I think you might not be doing it in the way the authors of the model suggest. Reading through Mancini & Marzocchi (2023, MM23), it appears that they impose a set of parameters fixed to specific values that you are not actually using (e.g., p=1.5 vs. your p=1.06, c=0.005 days vs. your c=0.04, b=1 vs. your b=0.85). Therefore, I believe you should estimate again the two free parameters while fixing the other six parameters from the MM23 table, then simulate again the simplETAS catalogs. Alternatively, you should clarify that you are mimicking the simplETAS approach, but with a different set of parameters (not desirable, though).

3) I am doubtful of how useful it is to plot all the 10k simulated catalogs in Figure 10, and what could be its take-home message? ETAS just does not divide earthquakes into 'mainshocks' and 'aftershocks', so maybe it is not surprising to see that the spatial distribution is different from Section 3.2? Also, assuming that the map in Figure 10 reflects the smaller square of Figure 9 (if I understand well), why is the imprint of the backgournd seismicity PDF missing in Figure 10? In other words, I would expect background events to be primarily placed where the bg-probability is larger (e.g., at the bottom-left side of the illustrated region) and then aftershocks to cluster all around, instead of the mostly homogeneous (with a few exceptions) distribution of earthquakes that is reported here. The difference between the estimated background seismicity spatial PDF presented in Figure 9 and the calculation sites and the potential seismic sources in and around Xichang (Figure 3) make the ETAS and the Omi-R-J model implementations very difficult to compare, even just conceptually. To fix this you might comment on the issue more thoroughly or try and use the calculation sites of Figure 3 as your background PDF to feed the ETAS simulations.

---

## Author Response (AR2)

Major points:

1) I would suggest moving the ETAS simulations part from the "Discussion" section to the earlier sections (maybe 3.2?) where you present the results. It looks strange to see additional modelling introduced and presented in a section where results should only be commented.

Reply: Thanks for your suggestion. Considering that Section 3.2 is a relatively independent section, we set a new section titled "5 Comparison with the ETAS Model" before "Discussion" to elaborate the ETAS simulations part.

2) I find the idea of using the "simplETAS" model appropriate. However I think you might not be doing it in the way the authors of the model suggest. Reading through Mancini & Marzocchi (2023, MM23), it appears that they impose a set of parameters fixed to specific values that you are not actually using (e.g., p=1.5 vs. your p=1.06, c=0.005 days vs. your c=0.04, b=1 vs. your b=0.85). Therefore, I believe you should estimate again the two free parameters while fixing the other six parameters from the MM23 table, then simulate again the simplETAS catalogs. Alternatively, you should clarify that you are mimicking the simplETAS approach, but with a different set of parameters (not desirable, though).

Reply: Thanks for your comments. We fixed the six parameters just as Mancini & Marzocchi (2023) did, and the estimated v and A were 23.8394 and 0.0212, respectively, as renewed in Table 4. We then used the new set of parameters to simulate 10000 catalogs for further comparison.

3) I am doubtful of how useful it is to plot all the 10k simulated catalogs in Figure 10, and what could be its take-home message? ETAS just does not divide earthquakes into 'mainshocks' and 'aftershocks', so maybe it is not surprising to see that the spatial distribution is different from Section 3.2? Also, assuming that the map in Figure 10 reflects the smaller square of Figure 9 (if I understand well), why is the imprint of the backgournd seismicity PDF missing in Figure 10? In other words, I would expect background events to be primarily placed where the bg-probability is larger (e.g., at the bottom-left side of the illustrated region) and then aftershocks to cluster all around,

instead of the mostly homogeneous (with a few exceptions) distribution of earthquakes that is reported here. The difference between the estimated background seismicity spatial PDF presented in Figure 9 and the calculation sites and the potential seismic sources in and around Xichang (Figure 3) make the ETAS and the Omi-R-J model implementations very difficult to compare, even just conceptually. To fix this you might comment on the issue more thoroughly or try and use the calculation sites of Figure 3 as your background PDF to feed the ETAS simulations.

Reply: Thanks for your comments. Due to the strong randomness of single simulated earthquake catalog, we simulated 10,000 sets of earthquake catalogs and stacked them so that we can observe the display of earthquake clusters effectively. In fact, when we enlarge the range of Figure 10, we can see an earthquake cluster at the bottom-left side both from the previous catalog and from the newly simulated catalog after parameter modification (please see the updated Figure 10).

The potential source models we employed to simulate earthquake catalogs in Section 3 comprehensively consider various data, including paleoearthquakes, historical earthquakes, seismogenic structures, stress-strain fields. These data help constrain the locations of earthquakes, especially those of high magnitude. However, it's important to note that the ETAS model is an empirical statistical model, relying on earthquake catalogs as its fundamental data. This distinction makes it challenging to draw direct comparisons between the two models. To address this limitation, it is essential for future research to explore the incorporation of more physics-based models to establish comparative bridges. However, this endeavor goes beyond the scope of the current study. We have included this comment in the revised manuscript.

Additionally, a minor oversight in Table 3 has been fixed.